# Immune dysregulation, apoptosis impairment, and enhanced seroreactivity to *Anisakis simplex* in Crohn's disease: interplay of IL-7/IL-7R signalling and CD132 deficiency

Carmen Cuéllar[1,2]/[+], Carolina Hurtado-Marcos[3], Elizabeth Valdivieso[3], Lucianna Vaccaro[3], Juan González-Fernández[1,2], Ana Isabel Jiménez[4], Jaume Pérez-Griera[5], Salvador Benlloch[6,7], Cirilo Amorós[6], Rafael Gil-Borrás[7], Rosa Sorando-Serra[8], María José Cano-Cano[8], Juan Carlos Andreu-Ballester[2,9]

[1]Complutense University of Madrid, Department of Microbiology and Parasitology, Madrid, Spain
[2]Complutense University of Madrid, Parasitic Immunobiology and Immunomodulation Research Group, Madrid, Spain
[3]CEU Universities, Universidad CEU San Pablo, Urbanización Montepríncipe, Boadilla del Monte, Spain
[4]Arnau de Vilanova Hospital, Biopathology Department, Valencia, Spain
[5]University Clinical Hospital, Laboratory Department, Valencia, Spain
[6]Arnau de Vilanova Hospital, Digestive Departament, Valencia, Spain
[7]CEU Universities, Cardenal Herrera, Valencia, Spain
[8]Arnau de Vilanova Hospital, Emergency Department, Valencia, Spain
[9]Foundation for the Promotion of Health and Biomedical Research in the Valencian Region, Benimaclet, Valencia, Spain

**BACKGROUND** In previous studies, we identified a deficiency of γδ T cells and an increased prevalence of anti-*Anisakis simplex* antibodies in patients with Crohn's disease (CD). Additionally, decreased gene expression of the interleukin 2 (IL-2) receptor γ subunit (CD132) was observed in tissues from CD patients.

**OBJECTIVE** To analyse the gene expression of IL-7 and its receptors in tissues from CD patients and to explore its relationship with anti-*A. simplex* antibodies.

**METHODS** 52 patients diagnosed with CD were compared with a control group of 52 healthy individuals. Peripheral blood samples were analysed to assess levels of anti-*A. simplex* antibodies and IL-7. In addition, intestinal tissue samples from 20 subjects in each group were examined to evaluate IL-7 gene expression, IL-7 protein levels, the IL-2 receptor γ subunit (CD132), the IL-7 receptor α subunit (CD127), and caspase-3 expression.

**FINDINGS** Anti-*A. simplex* antibody levels were elevated in patients with CD. Caspase-3 expression was significantly reduced in the tissues of CD patients with anti-*A. simplex* IgA, and this reduction extended to IgG and IgE in healthy individuals. A negative correlation was observed between caspase-3 levels and serum anti-*A. simplex* IgA, as well as IL-7 levels in the tissues of CD patients. In healthy subjects, tissue IL-7 levels were lower in those positive for anti-*A. simplex* IgA, while serum IL-7 levels were higher in individuals positive for anti-*A. simplex* IgG.

**MAIN CONCLUSIONS** This study revealed the interplay between IL-7 signalling, γδ T cell deficiency, and immune responses to *A. simplex* in CD. Our findings underscored a cause-effect relationship between CD132 deficiency, γδ T cell depletion, and defective mucosal immunity, which may drive both CD inflammation and susceptibility to parasitic infections like *A. simplex*.

Key words: Crohn's disease - anti-*Anisakis simplex* antibodies - IL-7 gene expression - IL-2 receptor subunit γ (CD 132) - caspase-3.

The aetiology of Crohn's disease (CD), a chronic inflammatory disorder of the gastrointestinal tract, remains elusive. However, its pathophysiology is believed to involve alterations in the innate immune system, particularly in genetically susceptible individuals exposed to specific pathogens.[1] Dysregulation of the innate immune system has been reported, including deficiencies in γδ T cells, which are predominantly located in mucosal membranes and serve as a critical first line of defence against pathogenic threats.[2]

The disease has also been linked to dysbiosis within the gut microbiota, characterised by an imbalance between beneficial microbes and pathogenic species. Emerging evidence highlights the role of newly identified intestinal microbes with pathogenic properties, termed "pathobionts," in the development and progression of CD. These pathobionts exhibit unique mechanisms that contribute to chronic inflammation, including escape of immune regulation and exacerbation of intestinal damage.[3]

+ Corresponding author: cuellarh@ucm.es | ⊙ https://orcid.org/0000-0001-7948-9889

The relationship between intestinal anisakiasis and CD has been explored in several studies.[4,5,6,7] *Anisakis simplex*, a nematode parasite, uses crustaceans and fish as intermediate hosts, with humans serving as accidental hosts. Human anisakiasis occurs upon ingestion of these larvae, leading to gastrointestinal and systemic infections often accompanied by allergic reactions, representing a significant public health concern.[8]

Studies have identified an association between specific anti-*A. simplex* antibodies and a deficiency in γδ T cells, suggesting that these cells play a critical role in the immune response against the parasite.[9] Recent research has demonstrated that patients with CD exhibit deficient gene expression of interleukin 2 (IL-2) receptor subunit γ (CD132), a component of the IL-7 receptor essential for γδ T cell stimulation and proliferation.[10] Furthermore, CD patients show reduced γδ T cell levels in tissues and peripheral blood, increased apoptosis of these cells, and decreased caspase-3 activity in the affected tissues. These findings highlight the potential immunological interplay between anisakiasis and the pathogenesis of CD. Taken together, these findings highlight the clinical relevance of IL-7R modulation in CD. The altered IL-7/IL-7R signalling pathway we describe, influenced by *A. simplex* immune responses, may contribute not only to defective γδ T cell homeostasis but also to disease persistence and progression. Since IL-7R expression (particularly the common γ-chain, CD132) has been proposed as a potential biomarker of treatment resistance and disease severity in CD,[11] our data suggest that anti-*A. simplex* antibodies and IL-7R dysfunction could serve as combined indicators of mucosal immune dysregulation. Moreover, therapeutic strategies aimed at restoring IL-7R signalling or mimicking the immunomodulatory effects observed in parasite exposure could open novel avenues for diagnosis and treatment in CD.

*Anisakis simplex* is of specific interest in CD due to overlapping epidemiological, clinical, and immunological features that suggest a possible role for this parasite in CD pathogenesis and presentation. Epidemiologically, anti-*Anisakis* antibodies have been reported at higher prevalence among CD patients than in healthy controls, with some studies showing specific immunoglobulins (notably IgA and IgG) detected in up to 29-44% of CD patients, which is disproportionately high compared to the healthy population.[5,12]

Clinically, intestinal anisakiasis and CD share overlapping symptoms such as abdominal pain and granulomatous inflammation, and *Anisakis* infection can mimic the presentation of CD, occasionally leading to diagnostic confusion and unnecessary interventions. There are documented cases in which *Anisakis* infection was initially misdiagnosed as CD based on clinical and histopathological findings.[12]

Immunologically, *A. simplex* infection stimulates a pronounced Th2-type immune response with increased levels of specific IgE, IgA, and IgG antibodies, as well as local eosinophilia, parameters that coincide with the immunological profile often observed in CD. In CD patients, the presence of anti-*Anisakis* IgA has been associated with higher CD activity indices, supporting a possible modulatory or exacerbating influence of the parasite's antigens on disease severity and mucosal immune activation.[5,12]

In summary, *A. simplex* represents a frequent, clinically relevant, and immunologically active coinfection in patients with CD, justifying its particular interest in the context of CD pathogenesis, diagnosis, and the broader understanding of host-parasite interactions in inflammatory bowel conditions.[5,12]

Our objective was to analyse the gene expression of IL-7 and its receptors in the tissues of patients with CD, with the aim of establishing a relationship between this cytokine and anti-*A. simplex* antibodies.

## SUBJECTS AND METHODS

*Study population* - We conducted a prospective case-control study to analyse the peripheral blood of 104 individuals, including 52 patients diagnosed with CD and 52 healthy controls. Additionally, we compared the tissue samples from 20 of these patients with those from 20 healthy subjects.

Intestinal tissue samples from CD patients were obtained via endoscopic or surgical biopsies, whereas control tissues from healthy subjects were collected during protocolised colon cancer screening programs showing normal findings. Participants were recruited at Arnau de Vilanova Hospital (Valencia, Spain), and patients and healthy controls were matched by sex and age (± 5 years). CD diagnosis adhered to the Lennard-Jones criteria, which integrate clinical, endoscopic, radiological, and histopathological features, including transmural inflammation, granulomas, and mucosal discontinuity.[13] Disease activity was assessed using the CD Activity Index (CDAI), calculated from weighted parameters, such as stool frequency, abdominal pain, and haematocrit levels.[14] The patients were stratified into three groups:

Newly diagnosed: active CD at presentation with no prior treatment or treatment initiation within ≤ 24 h.

Remission: sustained CDAI < 150 for ≥ 12 months.

Active disease: CDAI > 150.

Healthy controls excluded individuals with recent vaccinations (within three months), immunosuppressive therapies, immunodeficiencies, or autoimmune/inflammatory conditions.

*Tissue sampling of intestinal biopsies* - Three to five biopsy specimens from the ileum and colon were collected at 1-2 cm intervals in each subject. Select samples underwent fixation via immersion in 10% neutral-buffered formalin solution (pH 7.4) and were routinely processed through paraffin embedding for histological analysis. Tissue sections (4-5 μm thickness) were mounted on glass slides and stained with haematoxylin-eosin for microscopic evaluation.

Parallel samples were snap-frozen in liquid nitrogen without prior fixation to preserve the macromolecular integrity for subsequent cellular and molecular profiling. Frozen specimens underwent mechanical homogenisation in a dissociation medium containing phosphate-buffered saline (PBS, pH 7.4), 1% foetal bovine serum (FBS), 1 mM dithiothreitol (DTT), and 1 mM ethylenediaminetetraacetic acid (EDTA), followed by

incubation at 37ºC for 15 min. After centrifugation at 300 × g for 5 min, the cellular pellet was subjected to enzymatic digestion using 0.5 mg/mL collagenase type VIII (Sigma-Aldrich) in 5% FBS-supplemented medium, with continuous agitation at 37ºC for 30 min.

The resulting cell suspension was filtered through a 70-μm nylon mesh, pelleted by centrifugation, and immunolabeled with fluorochrome-conjugated antibodies targeting lineage-specific surface markers for flow cytometric analysis. Intestinal lymphocytes were isolated from healthy controls and CD patients for comparative functional studies.

*Determination of anti-A. simplex specific antibodies* - Enzyme-linked immunosorbent assay (ELISA) plates (Costar, Corning, NY, USA) were coated with larval antigen at a concentration of 10 μg/mL. Human serum samples, diluted 1:100 in PBS-Tween containing 0.1% bovine serum albumin (BSA), were added and incubated. Detection was performed using horseradish peroxidase (HRP)-conjugated goat anti-human IgM, IgG, or IgA antibodies (BioSource International, Camarillo, CA, USA). For IgE determination, serum samples were diluted 1:2 and incubated with a murine monoclonal antibody specific for the epsilon chain of human IgE (IgG1κ, clone E21A11; INGENASA, Madrid, Spain). Subsequently, a goat anti-mouse IgG1 (γ) HRP conjugate (Life Technologies, Grand Island, NY, USA) was used.

The participants were categorised into two groups based on their anti-*A. simplex* antibody levels for quantitative comparisons. Positive results were defined as optical density (OD) values exceeding the mean OD of the studied sera plus two standard deviations for each immunoglobulin type.[9]

*IL-7 in peripheral blood* - The concentration of serum interleukin-7 (IL-7) was quantified using an ELISA kit (Quantikine® HS ELISA, R&D Systems, Catalogue #: HS750) following the manufacturer's protocol.

*Cell isolation for the analysis of γδ and αβ T lymphocytes and apoptosis assessment* - Blood cell counts were performed using an automated haematology analyser (LH750; Beckman Coulter, Inc., Fullerton, CA, USA). To enrich the sample in mononuclear cells, density gradient centrifugation was performed on EDTA-anticoagulated blood using Lymphoprep™ (Palex Medical SA, Barcelona, Spain). Following two PBS washes, the collected cells were resuspended in 200 μL of binding buffer from the ANNEXIN V-FITC/7-AAD Kit (Beckman Coulter, Inc.), in the presence of calcium.

*Analysis of γδ and αβ T cells* - To assess the functional profile of γδ and αβ T lymphocytes in peripheral blood and intestinal tissue, flow cytometry was performed using the following monoclonal antibodies: Anti-TCR PAN αβ-PE (Catalogue #: B49177), Anti-TCR PAN γδ-PE and FITC (Catalogue #: 6607015), CD19-PC7 (Catalogue #: IM3628), CD56-PC7 and PE (Catalogue #: A21692; B36214), CD4-PC7 (Catalogue #: 737660), CD3-PC5 and ECD (Catalogue #: A07749; 6607013), CD8-PC7 and FITC (Catalogue #: 6607013), CD5-FITC (Catalogue #: IM0468U), and CD45-ECD (Catalogue #:

6607013) (Beckman Coulter). A total of 100,000 events were acquired using a multiparameter Navios flow cytometer (Beckman Coulter), and data were subsequently analysed with Kaluza software.

*Apoptosis evaluation* - Apoptosis in peripheral blood was assessed using the ANNEXIN V-FITC/7-AAD Kit (Beckman Coulter), which relies on Annexin V's affinity for phosphatidylserine exposed on apoptotic cells and the specificity of 7-amino-actinomycin D (7-AAD) for DNA guanine-cytosine base pairs. The assay was performed according to the manufacturer's instructions.

*Gene expression of IL-7, IL-7 receptor α (CD127), and IL-2 receptor γ subunit (CD132) in tissues. Reverse transcription real-time polymerase chain reaction (RT-qPCR)* - Tissue samples were homogenised in 1 mL of TRIzol® Reagent (Ambion, Life Technologies, Carlsbad, CA, USA) for RNA isolation, following the manufacturer's instructions. RNA concentration was determined using a GE NanoVue Spectrophotometer (GE Healthcare Life Sciences, Little Chalfont, UK).

Reverse transcription was performed using 1 μg of extracted RNA with a Thermo Scientific RevertAid H Minus First Strand cDNA Synthesis Kit (Thermo Fisher Scientific, Waltham, MA, USA. Catalogue Number K1621). The reaction was carried out by preparing the mixture with RNA template (1 μg), oligo dT (1 μL), and nuclease-free water up to 12 μL. Then, 5X Reaction Buffer (4 μL), RNase Inhibitor (20 U/μL, 1 μL), 10 mM dNTP mix (2 μL), RevertAid M-MuLV RT enzyme (200 U/μL, 1 μL), and water were added to a final volume of 20 μL. The mixture was incubated for 60 min at 42ºC, and the reaction was terminated by heating at 70ºC for 5 min. The product of the first strand cDNA synthesis can be used directly qPCR using SYBR Green Real Time PCR master mix Kit (Thermo Fisher Scientific, Waltham, MA, USA. Catalogue Number 4309153). Used to perform in a GeneAmp 5700 system (Applied Biosystems Foster City, CA, USA). For this, the reaction was carried out with 100 ng of cDNA from each sample, forward primer (100 nM, 1 μL), reverse primer (100 nM, 1 μL), 2X SYBR Green PCR Master Mix (25 μL), and water up to a final volume of 50 μL PCR conditions were as follows: 5 min at 94ºC; 40 cycles of 30 s at 94ºC, 30 s at 60ºC; 60 s at 72ºC. Glyceraldehyde 3-phosphate dehydrogenase (GAPDH) served as the internal reference gene. Gene expression level were calculated by the $2^{-\Delta\Delta CT}$ method. Primer sequences are listed in Supplementary data (Table A).

*IL-7 and caspase-3 protein expression. Western-blot* - Western blot analysis was performed using protein extracts from the intestinal biopsies. Protein purification was performed using the Trizol/Guanidine method with Trizol® Reagent (Ambion, Life Technologies) and Guanidine Hydrochloride (SIGMA), following the manufacturer's guidelines. The extracts were preserved in TNT buffer (20 mM Tris-HCl, pH 7.5, 0.2 M NaCl, 1% Triton X-100) and supplemented with a protease inhibitor cocktail (10%) (Roche, Vienna, Austria) immediately before use. Protein concentration was measured using the Pierce BCA Protein Assay Reagent (Thermo Fisher Scientific, Waltham, MA, USA).

A total of 20 µg of protein was separated by sodium dodecyl sulphate-polyacrylamide gel electrophoresis (SDS-PAGE) using 12% polyacrylamide gels. Proteins were subsequently transferred onto a nitrocellulose membrane using the Bio-Rad Mini Protean II system, following the manufacturer's recommendations (Bio-Rad, Hamburg, Germany).

Membranes were incubated overnight at 4ºC with the corresponding primary antibody [anti-IL-7 or anti-caspase-3; Supplementary data (Table B)] in TBS buffer (20 mM Tris-HCl, pH 7.5, and 150 mM NaCl) containing a 0.1% blocking agent. Excess primary antibody was removed by washing with TBS supplemented with 0.05% Tween-20, followed by incubation with a 1:1000 dilution of secondary antibody (Merck, Darmstadt, Germany). After additional washes, positive bands were visualised using an image digitiser with enhanced chemiluminescence (ECL) reagent (Amersham, Little Chalfont, UK) and quantified by densitometry using ImageJ software. Actin was used as an internal loading control.

*Statistical analysis* - The Mann-Whitney U test was employed to assess differences in medians between two independent groups of quantitative variables, as it is a robust non-parametric alternative to the t-test when data do not meet normality assumptions. Contingency tables were constructed to examine the relationship between anti-*A. simplex* antibodies, and other qualitative variables. For this purpose, chi-square tests and Fisher's exact tests were used to determine statistical associations.

Spearman's correlation coefficient was used for correlation analyses, comparing IL-7 gene expression, serum IL-7 levels, caspase-3 activity, and anti-*A. simplex* antibody levels. Graphical representations were generated using GraphPad Prism (version 8.0.0; GraphPad Software, San Diego, CA, USA). All statistical analyses were conducted using Statistical Package for Social Sciences (SPSS) version 19.0 (SPSS Inc., Chicago, IL, USA).

*Ethical approval* - This research was conducted in accordance with the recommendations of the Spanish Bioethics Committee, the Spanish legislation on Biomedical Research (Law 14/2007), and the Personal Data Protection Law (Law 3/2018 and European Law UE676/2018). The study was reviewed and approved by the Ethics and Research Committee of Arnau University Hospital of Vilanova-Lliria (Valencia, Spain). All participants, including patients and healthy controls, provided written informed consent for their participation, ensuring their anonymity throughout the study.

### RESULTS

Table shows the clinical characteristics of CD patients (n = 52).

*Anti-A. simplex antibodies in patients with CD and healthy subjects* - Fig. 1A illustrates the differences in anti-*A. simplex* antibody levels between CD patients and healthy controls, revealing significantly elevated levels of IgG and IgM in CD patients. Similarly, Fig. 1B highlights a higher prevalence of IgG and IgM positivity among CD patients. No significant variations in anti-*A. simplex* immunoglobulin levels were observed concerning sex, clinical settings, or Montreal classification.

*Values of IL-7 and its receptor caspase-3 in patients with CD and healthy subjects* - The levels of IL-7 gene expression and IL-7 protein in patients with CD were significantly elevated compared to healthy controls. Conversely, caspase-3 levels were reduced in the tissues of CD patients. The expression of the IL-2 receptor subunit γ (CD132) was markedly lower in CD tissues compared to healthy subjects. However, no significant differences were observed in the expression of IL-7 receptor α (CD127) or in tissue IL-7 levels. These findings are illustrated in Fig. 1C-D.

*Assessment of IL-7 and its receptor according to positivity of anti-A. simplex antibodies* - Fig. 2 illustrates the expression and distribution of IL-7 and related immune markers in CD patients, stratified by the presence or absence of anti-*A. simplex* antibodies. Panel A depicts IL-7 gene expression, while Panel B shows IL-7 protein levels in tissues. Serum IL-7 concentrations are presented in Panel C, and caspase-3 levels in tissues are shown in Panel D. Panels E and F display the expression of IL-2 receptor subunits γ (CD132) and α (CD127), respectively. Notably, a significant decrease in caspase-3 levels was observed in CD patients who were positive for IgA anti-*A. simplex* antibodies (Fig. 2D).

Figure 3 highlights findings in healthy individuals. Among these subjects, tissue IL-7 levels decreased in those positive for anti-*A. simplex* IgA antibodies, whereas serum IL-7 concentrations increased in individuals positive for anti-*A. simplex* IgG antibodies.

TABLE

Clinical characteristics of
Crohn's disease (CD) patients (n = 52)

|  |  | Mean (SD) |
|---|---|---|
| Age |  | 38.8 ± 11.6 |
| Harvey-Bradshaw Index |  | 7.4 ± 3.6 |
| Crohn's Disease Activity Index |  | 173.5 ± 116.5 |
|  |  | n (%) |
| Gender (Female) |  | 27 (51.9) |
| Clinical scenarios | New patient | 11 (21.2) |
|  | Remission | 14 (26.9) |
|  | Active disease | 27 (51.9) |
| Montreal age | A1 (< 16) | 3 (5.8) |
|  | A2 (17-40) | 32 (61.5) |
|  | A3 (> 40) | 16 (30.8) |
| Montreal location | L1 (Ileal) | 27 (51.9) |
|  | L2 (Colonic) | 4 (7.7) |
|  | L3 (Ileocolic) | 21 (40.4) |
| Montreal pattern | B1 (Inflammatory) | 22 (42.3) |
|  | B2 (Stenotic) | 17 (32.7) |
|  | B3 (Fistulising) | 13 (25.0) |

SD: standard deviation

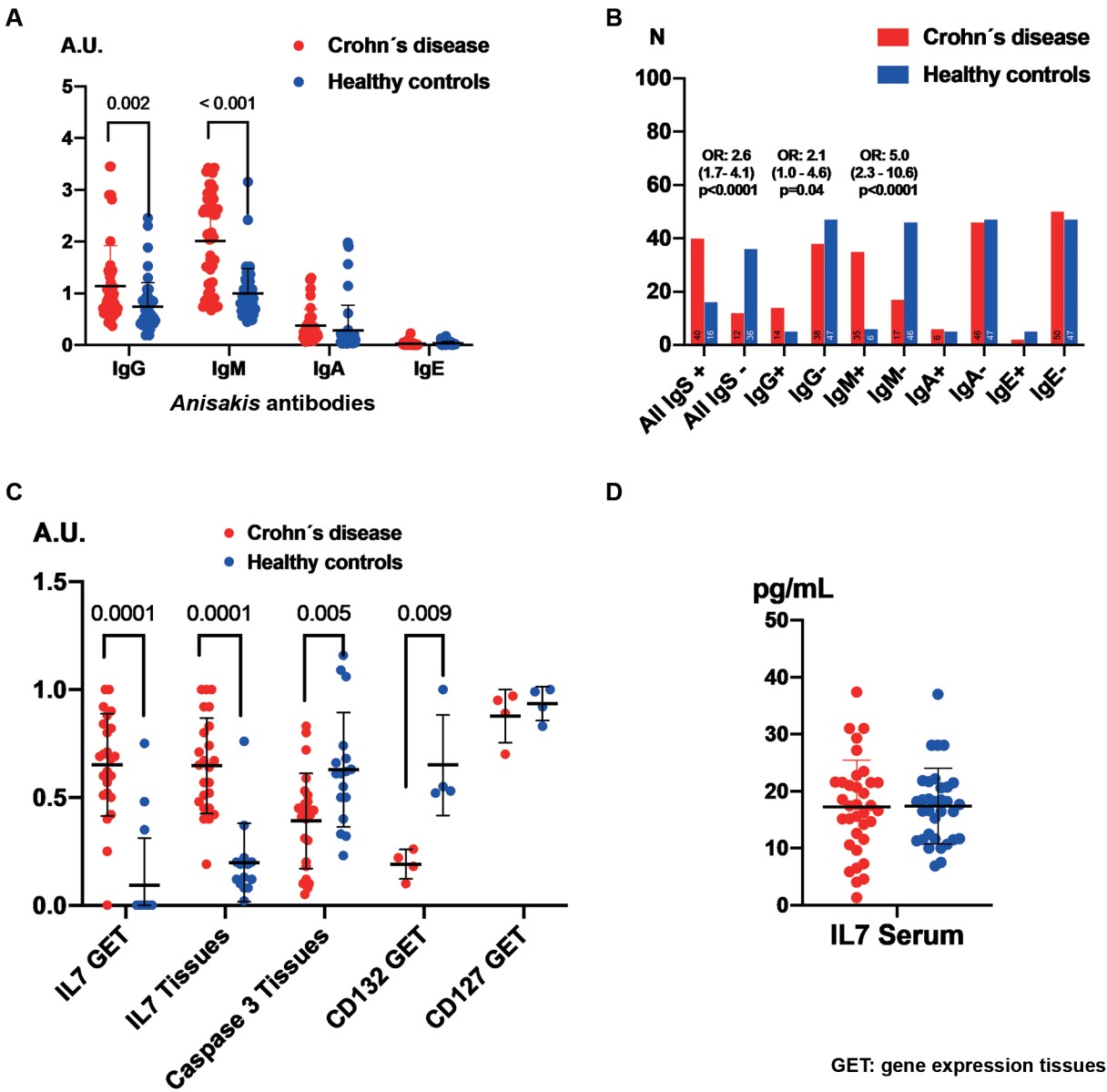

Fig. 1: the figure presents quantitative comparisons of anti-*Anisakis simplex* antibody levels, serum interleukin 7 (IL-7), and tissue levels of IL-7, its receptor, and caspase-3 between Crohn's disease (CD) patients and healthy controls. Panels A, C and D use the Mann-Whitney U test for continuous data, whereas Panel B displays categorical data analysed with Chi-square and Fisher's exact tests. Means are shown with standard deviations to indicate group variability.

*Correlations caspase-3 and IL-7 tissues with anti-A. simplex antibodies* - The relationship between IL-7, caspase-3, and anti-*A. simplex* antibody responses in tissues and serum can be summarised as follows.

A direct correlation existed between IL-7 gene expression and IL-7 cytokine levels in tissues for both CD patients and healthy individuals (Fig. 4A-B). In healthy subjects, tissue IL-7 levels were inversely correlated with serum IgA anti-*A. simplex* antibodies (Fig. 4D).

In CD patients, tissue caspase-3 expression was inversely related to both tissue IL-7 levels and serum IgA anti-*A. simplex* antibodies (Fig. 4C-D). Among healthy individuals, caspase-3 expression was inversely correlated with serum IgE anti-*A. simplex* antibodies (Fig. 4F).

In healthy individuals, serum IL-7 levels were positively correlated with serum IgG anti-*A. simplex* antibodies (Fig. 5A).

IL-2 receptor subunit γ (CD132) expression showed a positive correlation with serum IgM anti-*A. simplex* antibodies (Fig. 5B). IL-7Rα (CD127) expression was positively correlated with serum IgE anti-*A. simplex* antibodies (Fig. 5C) but negatively correlated with serum IgM anti-*A. simplex* antibodies (Fig. 5D).

All these correlations were statistically significant (p < 0.05).

*Caspase-3 protein expression and IL-7 receptor expression: IL-7 receptor α (CD127) and IL-2 receptor subunit γ (CD132) in tissues* - Supplementary data (Fig. 1)

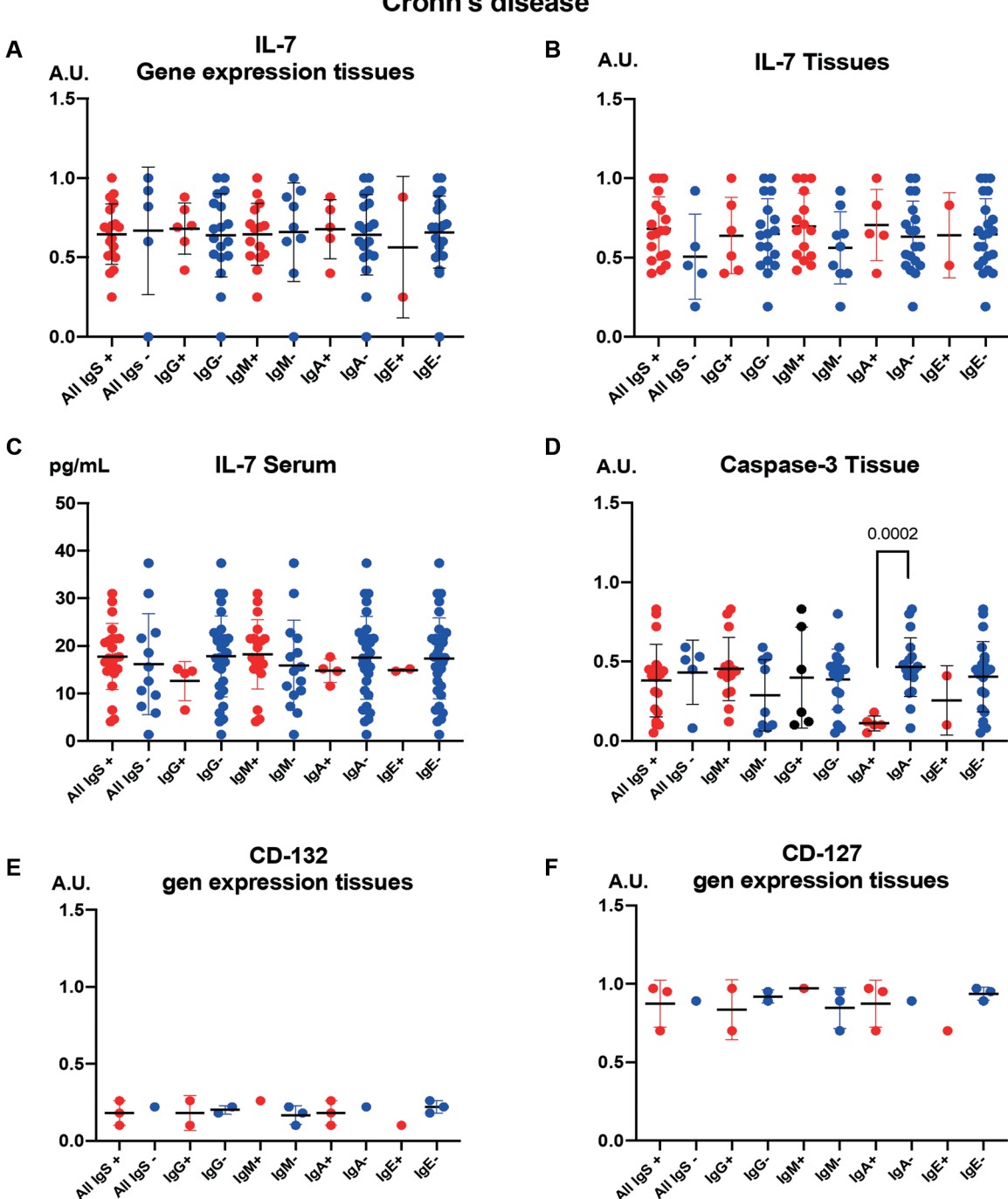

Fig. 2: the figure stratifies the immune parameters in Crohn's disease (CD) patients by the presence (+) or absence (-) of anti-*Anisakis simplex* antibodies. The figure is organised into panels showing: interleukin 7 (IL-7) gene expression (A); IL-7 protein levels in tissues (B); Serum IL-7 concentrations (C); Tissue caspase-3 levels (D); CD132 and CD127 receptor subunit expression (E and F). Statistical results are presented as means with double T-bars for standard deviation, with significance calculated via the Mann-Whitney U test.

shows IL-7 gene expression analysis (panel A), IL-7 protein expression (Panel B), CD127 and CD132 gene expression analysis (Panel C) and caspase-3 protein expression (Panel D). In patients with CD, tissue analysis demonstrated a significant increase in both IL-7 gene expression (p < 0.01) and IL-7 protein levels (p < 0.01) compared to healthy controls. Concurrently, caspase-3

titres in CD tissues showed a marked reduction relative to the control group (p < 0.01).

*Relationship between γδ T cells and apoptosis in CD patients compared to healthy controls* - Peripheral blood analysis revealed a significant reduction in γδ T cell subsets in CD patients versus controls [Supplementary data (Fig. 2 - Panel A)], with parallel decreases ob-

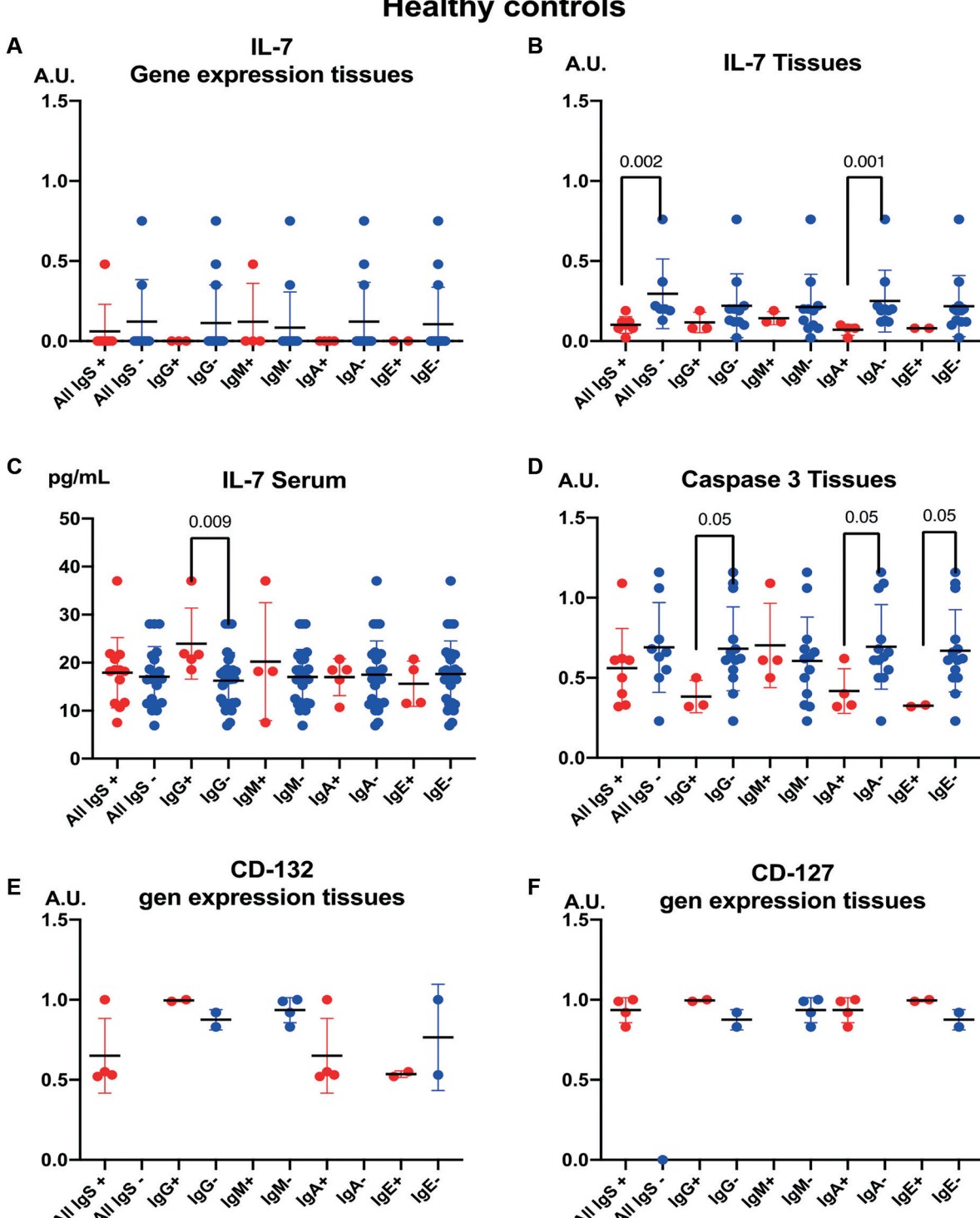

Fig. 3: the figure focuses on healthy controls, showing how tissue interleukin 7 (IL-7) and serum IL-7 vary according to the presence (+) or absence (-) of anti-*Anisakis simplex* antibodies. Data are depicted as means in arbitrary units, with standard deviations and Mann-Whitney U statistical comparisons.

served in intestinal tissues [Supplementary data (Fig. 2 - Panel B)]. Apoptotic activity showed elevated rates in CD3+CD56+ γδ T cell subsets within peripheral blood of CD patients compared to controls [Supplementary data (Fig. 2 - Panel C)].

Notably, antibody correlations exhibited an inverse relationship between anti-*Anisakis* IgE and IgA levels and apoptosis of CD3+CD56+ γδ T cells in peripheral blood [Supplementary data (Fig. 2 - Panels D, F)]. Conversely, a positive correlation existed between CD3+CD56+ γδ T

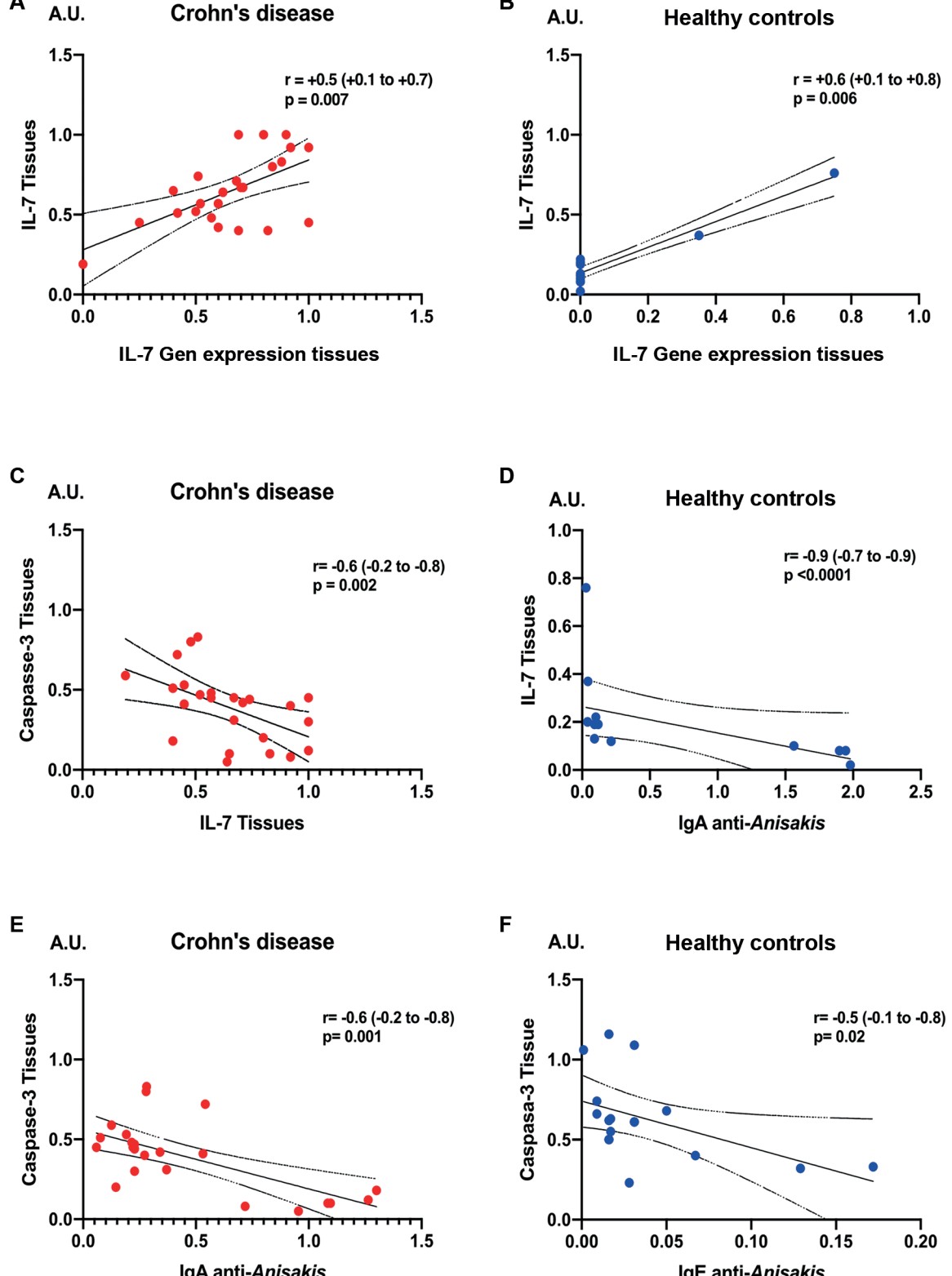

Fig. 4: the figure analyses correlations between tissue levels of caspase-3 and interleukin 7 (IL-7) as well as serum anti-*Anisakis simplex* antibodies using Spearman's rank correlation test, including 95% confidence intervals (CI). Both Crohn's disease (CD) patients and controls are included for direct statistical and biological correlations.

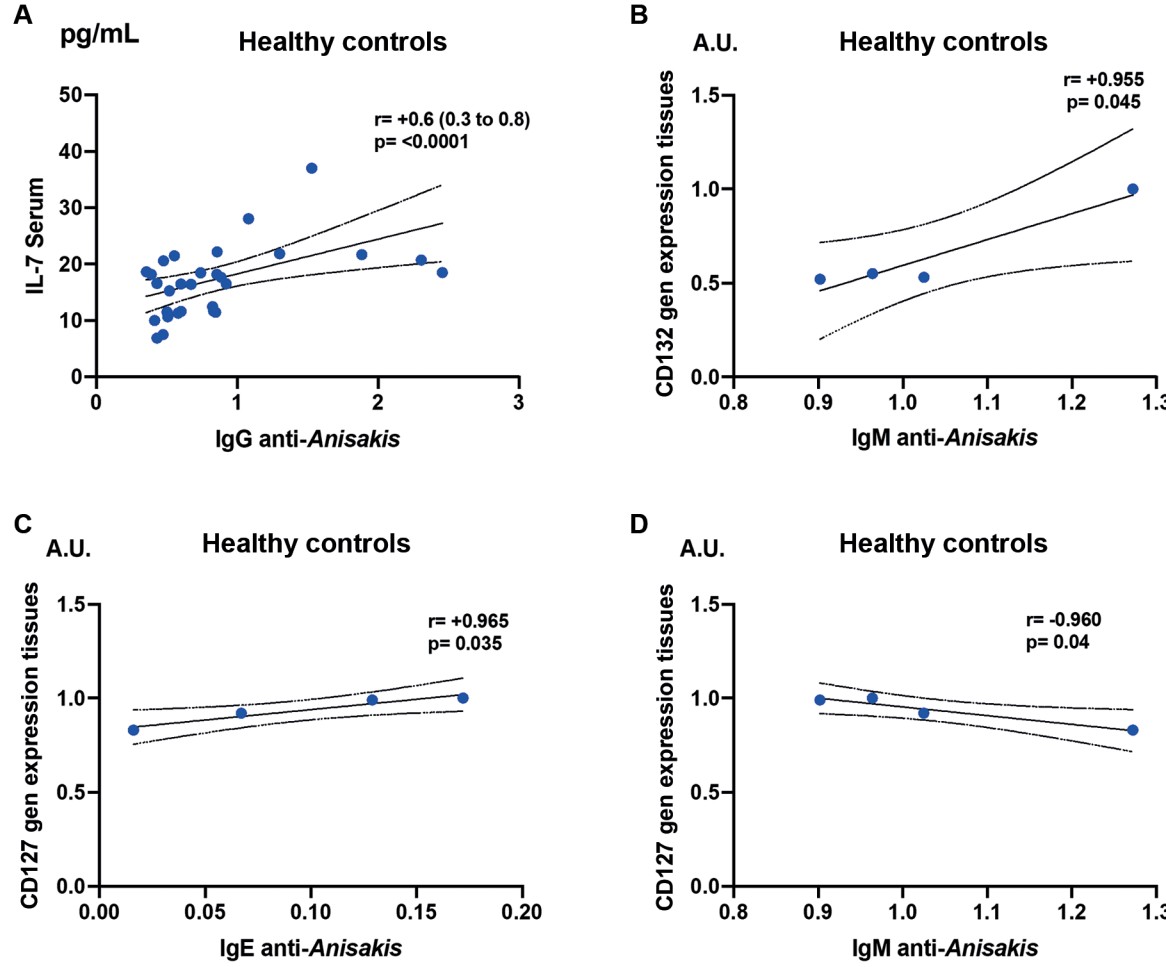

Fig. 5: the figure explores associations between serum interleukin 7 (IL-7) levels, tissue gene expression of the IL-7 receptor, and anti-*Anisakis simplex* antibody levels using Spearman's test. Subpanels detail positive or negative correlations with specific immunoglobulin classes, highlighting statistical significance.

cell numbers and anti-*Anisakis* IgA titres [Supplementary data (Fig. 2 - Panel E)]. No significant associations were detected between γδ T cell subsets in intestinal tissues and *Anisakis*-specific antibodies.

Analysis of apoptosis differences in CD8+ γδ T cell subsets in peripheral blood of CD patients based on the presence or absence of anti-*Anisakis* antibodies showed a significant decrease in IgG and IgA positive patients (p = 0.013 and 0,010, respectively) (data not shown).

### DISCUSSION

Previous investigations identified a significant depletion of γδ T cell populations alongside elevated serum levels of anti-*A. simplex* immunoglobulins in CD patients.[9,10,12] Recent genomic analyses further revealed diminished tissue expression of the IL-7R γ subunit (CD132), a critical component of the IL-7R heterodimer (IL-7Rα/CD127 and CD132).[10] This downregulation of CD132 correlated with mucosal immunodeficiency, potentially disrupting IL-7-mediated survival signals for γδ T cells despite elevated tissue IL-7 expression.[10,11] The current study aimed to characterise IL-7 and its receptor

dynamics in CD tissues, while investigating potential associations between these immunoregulatory pathways and anti-*A. simplex* antibody profiles, which may reflect compensatory immune responses to parasitic cofactors in disease pathogenesis.[5,9,12]

Our study confirmed an impaired IL-7R signalling in CD pathogenesis. As observed in previous studies, reduced gene expression of the common γ-chain receptor (CD132), a critical component of IL-7R, was observed in the intestinal tissues of patients with CD.[10,11] The relationship between IL-7R and the immune response to *A. simplex* can be inferred through the shared immunological mechanisms observed in parasitic infections and eosinophilic inflammation.[15,16] IL-7R is critical for eosinophil homeostasis in tissues.[17,18] In *A. simplex* infections, eosinophils dominate the inflammatory infiltrate during the larval penetration of the gastrointestinal mucosa. IL-7R-deficient mice showed reduced eosinophil reconstitution in the lungs, suggesting this receptor may similarly regulate eosinophilic responses to helminths.[18,19] *A. simplex* triggers a Th2-dominated response with IL-4, IL-5, and IL-13 production, which drives IgE synthesis

and eosinophil recruitment. IL-7R signalling influences T cell survival and cytokine profiles, potentially modulating Th2 differentiation. Impaired IL-7R function could alter this balance, affecting parasite clearance or hypersensitivity reactions. Both *A. simplex* infection and CD show mucosal TNF-α elevation and eosinophil activation. [11] The role of the IL-7R pathway in sustaining pathogenic T cells in CD may involve parallel mechanisms maintaining chronic inflammation in unresolved anisakiasis.

The results suggest that impaired IL-7R signalling in CD is associated with decreased expression of the common γ-chain receptor (CD132), leading to a deficiency in γδ T cells. This dysfunction may contribute to an inadequate immune response, increased susceptibility to infections, and chronic inflammation, which are critical factors in the pathogenesis of CD. These findings underscore the importance of IL-7R signalling in maintaining immune competence in affected individuals.

This deficiency disrupts the IL-7-mediated survival and maintenance of γδ T cells, contributing to their depletion in mucosal tissues. These findings suggest that restoring γδ IEL numbers or function could serve as a therapeutic strategy to prevent or mitigate CD development.[10,20]

We observed compensatory IL-7 overproduction with limited efficacy. CD tissues exhibited elevated IL-7 gene expression and protein levels, likely to counteract γδ T cell loss. However, serum IL-7 levels remained unchanged, suggesting defective paracrine/autocrine signalling due to CD132 deficiency.[10,21] In CD, there was a compensatory overproduction of IL-7 in response to diminished levels of γδ T cells. However, this increase in IL-7 did not translate into effective immune restoration due to the concurrent downregulation of the common γ-chain receptor (CD132). Consequently, despite higher IL-7 levels, the signalling through its receptor remained ineffective, leading to persistent immune deficiencies and heightened vulnerability to infections and chronic inflammation within the mucosal system. This failure in compensatory mechanisms highlights a critical aspect of the dysregulated immune environment in CD.

The results showed a caspase-3 reduction and apoptosis dysregulation. Lower caspase-3 levels in CD tissues were inversely correlated with tissue IL-7 and serum levels of anti-*A. simplex* IgA, indicating impaired apoptosis regulation. This aligns with IL-7's role in suppressing γδ T cells.[10,20] There was a notable decrease in caspase-3 expression in the tissues of CD patients, which correlated with the dysregulation of apoptosis in T cells, particularly γδ T cells. This lower caspase-3 level suggests a reduced apoptotic signal, potentially allowing for increased survival of T cells despite underlying immunodeficiency. However, this aberration in apoptosis does not rectify γδ T cell deficiency, contributing to chronic inflammation and impaired immune responses. These findings emphasise the complex interplay between reduced apoptotic signalling and maintenance of T cell homeostasis in the pathophysiology of CD. Caspase-3, a major apoptotic effector, plays a crucial role in the regulation of the immune response. Its reduction can lead to enhanced cytokine signalling and inflammation, which may contribute to the host response against *A. simplex*.[22] Likewise,

*A. simplex* produces an apoptosis-inducing protein that can trigger apoptosis in mammalian cells through two mechanisms: $H_2O_2$ production and L-lysine deprivation. Both mechanisms involve the release of cytochrome c and the activation of caspase-9.[23] Understanding caspase-3's role in apoptosis and immune regulation could provide insights into how *A. simplex* might evade or modulate host immune responses, a common strategy employed by parasites.[24]

Crohn's disease patients exhibited significantly elevated levels of anti-*A. simplex* IgG and IgM antibodies compared to healthy controls. A higher prevalence of IgG and IgM positivity was also observed among CD patients. No significant differences in anti-*A. simplex* immunoglobulin levels were found when analysed by sex, clinical settings, or Montreal classification. The association between anti-*A. simplex* antibodies and immunodeficiencies have been demonstrated previously. Furthermore, our study suggests that anti-*A. simplex* antibodies could serve as markers for disease progression risk in CD. CD patients showed elevated anti-*A. simplex* IgG, IgM, and IgA antibodies, with IgA levels being linked to higher disease activity.[9,12] The deficiency of γδ T cells, which is critical for mucosal surveillance, likely creates an immunosuppressed state, permitting *A. simplex* antigen persistence and chronic antibody production.[5,9,12]

We observed divergent IL-7 responses in CD versus healthy subjects. In healthy subjects, serum IL-7 increased with anti-*A. simplex* IgG, suggesting a functional immune response. Conversely, CD patients exhibited reduced tissue IL-7 in IgA-positive cases, reflecting dysregulated compensatory mechanisms.[10,12] The results highlight the divergent responses of IL-7 in CD patients compared to healthy controls. In CD, there was an increase in IL-7 gene expression and protein levels in intestinal tissues, indicating an attempt to compensate for the observed deficiency of γδ T cells. In contrast, serum IL-7 levels in CD patients remain comparable to those in healthy individuals, suggesting a potential disruption in the paracrine and autocrine functions of IL-7. This discrepancy points to a pathological mechanism where despite elevated locally produced IL-7, its effectiveness is compromised due to the downregulated expression of the common γ-chain receptor (CD132). This dysfunctional signalling may contribute to persistent immune deficits in CD, highlighting the complexity of IL-7's role in inflammation and immune regulation in affected individuals. CD patients had a higher prevalence of antibodies against *A. simplex* compared to healthy controls. This suggests an altered immune response in CD, which could be related to the divergent IL-7 responses observed in CD patients versus healthy subjects. IgA antibodies against *A. simplex* were associated with higher CD activity index. This correlation between antibody levels and disease activity might parallel potential relationships between IL-7 responses and CD severity.

Our results highlight CD132 and IL-7R signalling as potential biomarkers for CD progression and treatment resistance.[11] Targeting this pathway could restore γδ T cell function and mitigate *A. simplex*-associated immune dysregulation.[11,20]

These findings underscore a cause-effect relationship between CD132 deficiency, γδ T cell depletion, and defective mucosal immunity, which may drive both CD inflammation and susceptibility to parasitic infections like *A. simplex*.

We are aware that the study acknowledges several limitations that could impact on the validity and generalisability of its findings. The small cohort size limits statistical power, particularly for subgroup analyses, and expanding the sample size would enhance generalisability and allow for demographic stratification. While correlations between *A. simplex* infection and IL-7R expression changes are identified, causation remains unestablished. Longitudinal studies monitoring immune responses over time are needed to clarify causality and disease progression. Potential confounders like diet, concurrent infections, and treatments may skew results, necessitating detailed evaluations to isolate the role of *A. simplex*. Reliance on self-reported symptoms via the CDAI introduces bias; incorporating objective biomarkers such as faecal calprotectin or C-reactive protein could improve reliability. Additionally, mechanisms underlying *A. simplex* effects on IL-7R expression remain unexplored, requiring functional assays and *in vitro* studies for deeper insights. Addressing these limitations in future research would strengthen findings and understanding of immune modulation in CD.

In conclusion, the study revealed several key aspects regarding the interplay between IL-7 signalling, γδ T cell deficiency, and immune responses to *A. simplex* in CD.

Reduced gene expression of the common γ-chain receptor (CD132), a critical component of the IL-7R, was observed in CD patients' intestinal tissues. This deficiency disrupts IL-7-mediated survival and maintenance of γδ T cells, contributing to their depletion in mucosal tissues.

Crohn's disease tissues exhibited elevated IL-7 gene expression and protein levels, likely attempting to counteract γδ T cell loss. However, serum IL-7 levels remained unchanged, suggesting defective paracrine/autocrine signalling due to CD132 deficiency.

Lower caspase-3 levels in CD tissues correlated inversely with tissue IL-7 and serum anti-*A. simplex* IgA, indicating impaired apoptosis regulation. This aligns with IL-7's role in suppressing γδ T cell apoptosis.

Crohn's disease patients showed elevated anti-*A. simplex* IgG, IgM, and IgA antibodies, with IgA levels linked to higher disease activity. The deficiency in γδ T cells, critical for mucosal surveillance, likely creates an immunosuppressed state, permitting *A. simplex* antigen persistence and chronic antibody production.

In healthy subjects, serum IL-7 increased with anti-*A. simplex* IgG, suggesting a functional immune response. Conversely, CD patients exhibited reduced tissue IL-7 in IgA-positive cases, reflecting dysregulated compensatory mechanisms.

The study highlights CD132 and IL-7R signalling as potential biomarkers for CD progression and treatment resistance. Targeting this pathway could restore γδ T cell function and mitigate *A. simplex*-associated immune dysregulation.

These findings highlight the intricate interactions between cytokines, apoptosis regulators, and immune responses in both healthy individuals and those with CD.

## AUTHORS' CONTRIBUTION

CC, JCA-B and CH - conceptualisation; JCA-B - formal analysis; CC and JCA-B - funding acquisition; JP-G, SB, CA, RG-B, RS-S and MJC-C - investigation collecting data and recruitment of patients; CC and JG-F - investigation on IL-7 and anti-*Anisakis simplex* antibodies; CH, EV and LV - investigation on protein expression in tissues and investigation on tissue sampling; CC, JCA-B and CH - validation, visualisation, writing the original draft; CC, JG-F, JCA-B and CH - writing review and editing. The authors declare that they have no conflict of interest.

## DATA AVAILABILITY

The contents underlying the research text are included in the manuscript.

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

# OPEN PEER REVIEW

Memórias do IOC thanks the anonymous reviewers for their contribution to the peer review of this work.

## FIRST REVIEW ROUND

REVIEWERS' COMMENTS

### REVIEWER #1

The submitted manuscript explores the modulation of interleukin-7 receptor (IL-7R) gene expression in intestinal tissues of patients with Crohn's disease (CD), with a particular focus on the presence of anti-Anisakis simplex antibodies. The authors aim to elucidate the potential interplay between defective IL-7 signaling, γδ T cell deficiency, and immune responses to parasitic antigens in the context of CD pathogenesis.

While the topic is both timely and scientifically relevant—especially given the growing interest in host–parasite interactions and mucosal immunoregulation—the manuscript suffers from several critical weaknesses. These include redundancy in the introduction, insufficient methodological detail, speculative interpretation of correlative data, and the absence of mechanistic validation. Without thorough revision and clarification of key elements, the study falls short of the standards required for publication.

Introduction

1. Terminology: anisakiasis - infection by nematodes of the Anisakis genus (typically Anisakis simplex); anisakiosis - infection by nematodes of the Anisakidae family, which includes Anisakis, Pseudoterranova, and others. Based on that authors in my opinion should use term – anisakiasis.

2. The potential clinical importance of IL-7R modulation in CD (e.g., for diagnosis or therapy) is not mentioned. This would help justify the rationale for the study.

3. The role of betadelta T cells in anisakiasis: the Benet-Campos 2017 were studied patients with CD, not people only infected with parasitic nematodes and based on this paper it cannot be stated that; better citation would be Zamora et al. 10.1515/ap-2017-0011

4. Some references (e.g., del Pozo 1999) are more than two decades old. While historical studies are relevant, the inclusion of more recent literature would enhance the contemporary relevance of the introduction.

5. Insufficient justification for A. simplex-CD connection: although previous studies are cited, the authors do not sufficiently explain why this parasite is of specific interest in CD or how frequent or clinically relevant this coinfection is in the patients with Crohn's disease.

Materials and Methods

1. There is no explanation of how the sample size was determined or whether the study is statistically powered to detect meaningful differences—especially in subgroup comparisons. Authors should provide explanation about the tested group e.g. only that number of patients were available? then authors should provide sample size justification or power analysis.

2. Classification of anti-A. simplex antibody positivity based on OD > mean + 2 SD is not clearly referenced as a validated approach. The diagnostic performance of this cutoff should be addressed (e.g. serial dilutions) or specific methodology described in more detail based on cited references.

3. IL-7 ELISA kit catalog number is missing.

4. While CD patients are grouped by activity status (new, remission, active), the distribution of IL-7-related parameters across these clinical strata is not described, nor is the relationship with other tested parameters fully explored. Maybe supplementary file with the results divided into these three strata would be beneficial?

5. Antibodies used for analysis of T cells should also have catalog numbers provided.

6. What was the method to calculate gene expression, Livak or Pfaffl? Please provide detailed information in this matter.

7. Real time PCR and Western in M&M should be divided into two separate paragraphs. Add detailed information about real-time PCR (mix concentrations, temperature profile etc.)

8. In methodology authors stated that we're analyzing Il-7 and caspase-3 using Western Blot. However, I could not find blot image for IL-7. Authors should also provide whole blot images in supplementary files for reference.

9. Although actin is listed in the supplement, the main methods text should mention it clearly as a control and specify whether it was validated across all blots shown.

Results

1. Results are only qualitatively described (e.g., "higher", "lower") without reporting exact values. Including exact statistics and effect sizes would improve transparency and reproducibility. Moreover, the results are described superficially. The authors merely indicate what is presented in each panel and figure, without providing

specific comparisons, values, or interpretation of the graphs. This section should be expanded to highlight the most statistically significant findings, as well as those that are not significant, in order to provide a thorough explanation of data-rich figures—such as Figure 1 or 2.

2. Figure captions should be described in more detail. In the Fig 1 the results of IL7 ELISA should be marked as panel D.

3. In the fig 2/3 caption should be added that + is presence and – is absence.

4. The authors describe correlations between IL-7, caspase-3, and anti-A. simplex antibodies without referring to exact values of R. Please provide these data in the text.

5. Suppl. Fig. 2 is important and in my opinion, if possible, should be in the main text of the manuscript as Fig. 6.

Discussion

1. In the discussion section (page 11) the second paragraph seems to not be supported with significant citations.

2. AIP (apoptosis-inducing protein) is a protein purified and cloned from Chub mackerel infected with Anisakis simplex. Murakwa et al did not state that this is parasitic-origin protein. Authors should rewrite the sentence about this, due to that, their statement is not corresponding to that of Murakawa (page 12).

3. The conclusion highlights CD132 and IL-7R signaling as potential therapeutic targets in CD. However, this is speculative, as no intervention or in vitro modulation experiments were conducted to support this claim. Is this pathway relevant only in the context of CD combined with A. simplex-associated immune response?

Moroeover, the current title appears too narrow in relation to the breadth and complexity of the data presented. The manuscript addresses not only IL-7 receptor gene expression but also explores IL-7 protein levels, CD132 and CD127 expression, caspase-3 activity, apoptosis of γδ T cells, and multiple correlations with Anisakis simplex-specific antibodies in both Crohn's disease patients and healthy controls. Given the broader focus on immune dysregulation, apoptosis, and mucosal immunity, I recommend rephrasing the title to more accurately capture the full scope of the study. A more inclusive and descriptive title would improve discoverability and better reflect the manuscript's contribution to the field.

This manuscript provides several novel and valuable findings within the context of immune dysregulation in Crohn's disease, particularly in relation to IL-7 signaling, γδ T cell homeostasis, and seroreactivity to Anisakis simplex.

However, to meet the standards of scientific clarity and rigor required for publication, the manuscript requires substantial revisions—particularly in the areas of data interpretation, methodological detail, and presentation. Once these issues are adequately addressed, the study could be suitable for publication.

**AUTHORS' RESPONSE TO THE REVIEWERS**

Dear Editor,

We thank Reviewer 1 for their thoughtful and detailed assessment of our manuscript, which addresses the modulation of interleukin-7 receptor gene expression by Anisakis simplex in tissues from Crohn's disease patients. Below, we respond point-by-point to every raised issue, specifying modifications made and providing scientific rationale where specific requests present considerable methodological, technical, or interpretative challenges.

Reviewer Points & Author Responses
REVIEWER COMMENTS:
The submitted manuscript explores the modulation of interleukin-7 receptor (IL-7R) gene expression in intestinal tissues of patients with Crohn's disease (CD), with a particular focus on the presence of anti-Anisakis simplex antibodies. The authors aim to elucidate the potential interplay between defective IL-7 signaling, γδ T cell deficiency, and immune responses to parasitic antigens in the context of CD pathogenesis.

While the topic is both timely and scientifically relevant—especially given the growing interest in host–parasite interactions and mucosal immunoregulation—the manuscript suffers from several critical weaknesses. These include redundancy in the introduction, insufficient methodological detail, speculative interpretation of correlative data, and the absence of mechanistic validation. Without thorough revision and clarification of key elements, the study falls short of the standards required for publication.

Introduction
1. Terminology (anisakiasis vs anisakiosis)
Terminology: anisakiasis - infection by nematodes of the Anisakis genus (typically Anisakis simplex); anisakiosis - infection by nematodes of the Anisakidae family, which includes Anisakis, Pseudoterranova, and others. Based on that authors in my opinion should use term –anisakiasis.

REPLY: Regarding the terminology, we respectfully maintain the use of "anisakiosis" and not "anisakiasis" throughout the manuscript. This decision is based on the Standardized Nomenclature of Animal Parasitic Diseases

(SNOAPAD), first published in 1988 by the World Association for the Advancement of Veterinary Parasitology (WAAVP). SNOAPAD's central guideline is to form disease names from the taxonomic name of the parasite, combined with the suffix "-osis," to ensure clearer scientific communication and facilitate data retrieval. In the case of infections caused by Anisakis spp., the recommended term is "anisakiosis" (Kassai, T., Cordero del Campillo, M., Euzeby, J., Gaafar, S., Hiepe, T., & Himonas, C. A. (1988). Standardized nomenclature of animal parasitic diseases (SNOAPAD). Veterinary Parasitology, 29(4), 299–326. https://doi.org/10.1016/0304-4017(88)90148-3). For these reasons and in line with current international guidelines, we believe "anisakiosis" is the most appropriate term for this context.

2. Clinical importance of IL-7R modulation in CD

The potential clinical importance of IL-7R modulation in CD (e.g., for diagnosis or therapy) is not mentioned. This would help justify the rationale for the study.

REPLY: The revised introduction now specifically discusses the diagnostic and therapeutic relevance of IL-7R modulation in Crohn's disease and how our study contributes to this field. This sentence has been included: "Taken together, these findings highlight the clinical relevance of IL-7R modulation in Crohn's disease. The altered IL-7/IL-7R signaling pathway we describe, influenced by Anisakis simplex immune responses, may contribute not only to defective $\gamma\delta$ T cell homeostasis but also to disease persistence and progression. Since IL-7R expression (particularly the common $\gamma$-chain, CD132) has been proposed as a potential biomarker of treatment resistance and disease severity in CD (1), our data suggest that anti-A. simplex antibodies and IL-7R dysfunction could serve as combined indicators of mucosal immune dysregulation. Moreover, therapeutic strategies aimed at restoring IL-7R signaling or mimicking the immunomodulatory effects observed in parasite exposure could open novel avenues for diagnosis and treatment in Crohn's disease."

1.Belarif L, Danger R, Kermarrec L, Nerrière-Daguin V, Pengam S, Durand T, Mary C, Kerdreux E, Gauttier V, Kucik A, Thepenier V, Martin JC, Chang C, Rahman A, Guen NS, Braudeau C, Abidi A, David G, Malard F, Takoudju C, Martinet B, Gérard N, Neveu I, Neunlist M, Coron E, MacDonald TT, Desreumaux P, Mai HL, Le Bas-Bernardet S, Mosnier JF, Merad M, Josien R, Brouard S, Soulillou JP, Blancho G, Bourreille A, Naveilhan P, Vanhove B, Poirier N. IL-7 receptor influences anti-TNF responsiveness and T cell gut homing in inflammatory bowel disease. J Clin Invest. 2019 Apr 2;129(5):1910-1925. doi: 10.1172/JCI121668. PMID: 30939120; PMCID: PMC6486337.

3. Role of $\gamma\delta$ T cells and citation accuracy

The role of beta delta T cells in anisakiasis: the Benet-Campos 2017 were studied patients with CD, not people only infected with parasitic nematodes and based on this paper it cannot be stated that; better citation would be Zamora et al. 10.1515/ap-2017-0011

REPLY: We appreciate the reviewer's suggestion to cite Zamora et al. (2017) in relation to the role of $\gamma\delta$ T cells in anisakiosis. However, Benet-Campos et al. (2017) is a more suitable reference for our study, which specifically investigates interleukin-7 receptor gene expression and immune responses in Crohn's disease patients. The Benet-Campos paper directly examines both anti-Anisakis simplex antibodies and the relationship with $\alpha\beta$ and $\gamma\delta$ T cell subpopulations in Crohn's disease patients, providing context that closely aligns with our research focus. Zamora et al. (2017), by contrast, presents data on healthy subjects rather than Crohn's disease patients. Given that our manuscript addresses immunological modulation by Anisakis simplex within the Crohn's disease setting, it is scientifically appropriate to maintain the Benet-Campos et al. (2017) reference as it offers the most relevant evidence for the patient cohort under study.

4. Update of references

Some references (e.g., del Pozo 1999) are more than two decades old. While historical studies are relevant, the inclusion of more recent literature would enhance the contemporary relevance of the introduction.

REPLY: An additional and owned recent reference has been incorporated (de la Hoz-Martín, M. P., González-Fernández, J., Andreu-Ballester, J. C., Hoivik, M. L., Ricanek, P., Bruland, T., Sandvik, A. K., Cuéllar, C., & Catalán-Serra, I. (2025). Prevalence of Anti-Anisakis simplex Antibodies in a Cohort of Patients with Inflammatory Bowel Disease in Norway. Pathogens (Basel, Switzerland), 14(8), 769. https://doi.org/10.3390/pathogens14080769), preserving relevant historical context but improving the contemporary basis for our assertions.

5. Justification for A. simplex focus in CD

Insufficient justification for A. simplex-CD connection: although previous studies are cited, the authors do not sufficiently explain why this parasite is of specific interest in CD or how frequent or clinically relevant this coinfection is in the patients with Crohn's disease.

REPLY: We have expanded the rationale in the introduction, highlighting epidemiological, clinical, and immunological connections between A. simplex infection and Crohn's disease pathogenesis. The following paragraphs have been included: "A. simplex is of specific interest in CD due to overlapping epidemiological, clinical, and immunological features that suggest a possible role for this parasite in CD pathogenesis and presentation. Epidemiologically, anti-Anisakis antibodies have been reported at higher prevalence among CD

patients than in healthy controls, with some studies showing specific immunoglobulins (notably IgA and IgG) detected in up to 29-44% of CD patients, which is disproportionately high compared to the healthy population (Guillén-Bueno et al., 1999; Gutiérrez & Cuéllar, 2002).

Clinically, intestinal anisakiosis and CD share overlapping symptoms such as abdominal pain and granulomatous inflammation, and Anisakis infection can mimic the presentation of CD, occasionally leading to diagnostic confusion and unnecessary interventions. There are documented cases in which Anisakis infection was initially misdiagnosed as CD based on clinical and histopathological findings (Guillén-Bueno et al., 1999).

Immunologically, A. simplex infection stimulates a pronounced Th2-type immune response with increased levels of specific IgE, IgA, and IgG antibodies, as well as local eosinophilia, parameters that coincide with the immunological profile often observed in CD. In CD patients, the presence of anti-Anisakis IgA has been associated with higher CD activity indices, supporting a possible modulatory or exacerbating influence of the parasite's antigens on disease severity and mucosal immune activation (Guillén-Bueno et al., 1999; Gutiérrez & Cuéllar, 2002).

In summary, A. simplex represents a frequent, clinically relevant, and immunologically active coinfection in patients with CD, justifying its particular interest in the context of CD pathogenesis, diagnosis, and the broader understanding of host-parasite interactions in inflammatory bowel conditions (Guillén-Bueno et al., 1999; Gutiérez & Cuéllar, 2002)."

Materials and Methods

1. Sample size explanation/statistical power

There is no explanation of how the sample size was determined or whether the study is statistically powered to detect meaningful differences—especially in subgroup comparisons. Authors should provide explanation about the tested group e.g. only that number of patients were available? then authors should provide sample size justification or power analysis.

REPLY: The sample size (52 subjects) for a disease with such a low prevalence is one of the highest compared to other studies of this disease, and even more so given the statistical significance found (0.005-0.0001 - Figure 1), which provides a very powerful value. Furthermore, it far exceeds the commonly accepted threshold in biomedical studies to ensure the validity of parametric tests. In any case, the minimum sample size necessary to estimate an expected proportion of 10% based on the expected value and the accepted absolute error of 10% (or desired precision): the sample size would be 35 subjects, so we far exceed this figure. The logistical effort (recruiting 52 patients with this disease is difficult and time-consuming) and budgetary costs must also be considered. Expanding statistical power by increasing cohort size is not feasible for this study due to recruitment challenges.

2. Antibody positivity cut-off and diagnostic performance

Classification of anti-A. simplex antibody positivity based on OD > mean + 2 SD is not clearly referenced as a validated approach. The diagnostic performance of this cutoff should be addressed (e.g. serial dilutions) or specific methodology described in more detail based on cited references.

REPLY: The use of the mean optical density (OD) plus two standard deviations (mean+2SD) as a positivity cutoff for anti-Anisakis antibodies is a widely accepted and validated approach in serological studies of parasitic infections, including anisakiosis. This statistical criterion is recommended when a true gold standard is unavailable, as it ensures high specificity by distinguishing between background reactivity in healthy controls and true positive responses. Multiple published studies on anti-Anisakis ELISA utilize similar cut-off determinations, supporting the validity of this approach in the absence of comprehensive serial dilution data or reference standards (Anadón AM, Rodríguez E, Gárate MT, Cuéllar C, Romarís F, Chivato T, Rodero M, González-Díaz H, Ubeira FM. Diagnosing human anisakiasis: recombinant Ani s 1 and Ani s 7 allergens versus the UniCAP 100 fluorescence enzyme immunoassay. Clin Vaccine Immunol. 2010 Apr;17(4):496-502. doi: 10.1128/CVI.00443-09. Epub 2010 Jan 27. PMID: 20107002; PMCID: PMC2849323; de Las Vecillas L, Muñoz-Cacho P, López-Hoyos M, Monttecchiani V, Martínez-Sernández V, Ubeira FM, Rodríguez-Fernández F. Analysis of Ani s 7 and Ani s 1 allergens as biomarkers of sensitization and allergy severity in human anisakiasis. Sci Rep. 2020 Jul 9;10(1):11275. doi: 10.1038/s41598-020-67786-w. Erratum in: Sci Rep. 2020 Oct 28;10(1):18808. doi: 10.1038/s41598-020-75954-1. PMID: 32647149; PMCID: PMC7347943; Martínez-Aranguren, R. M., Gamboa, P. M., García-Lirio, E., Asturias, J., Goikoetxea, M. J., & Sanz, M. L. (2014). In vivo and in vitro testing with rAni s 1 can facilitate diagnosis of Anisakis simplex allergy. Journal of investigational allergology & clinical immunology, 24(6), 431–438).

As more extensive performance characterization, such as calculation of ROC curves or titration-based specificity/sensitivity, is not feasible without serial dilution data, using the mean+2SD cutoff provides a robust, evidence-based threshold that aligns with best practices in the field (Martínez-Aranguren, R. M., Gamboa, P. M., García-Lirio, E., Asturias, J., Goikoetxea, M. J., & Sanz, M. L. (2014). In vivo and in vitro testing with rAni s 1 can facilitate diagnosis of Anisakis simplex allergy. Journal of investigational allergology & clinical immunology, 24(6), 431–438.; Anadón AM, Rodríguez E, Gárate MT, Cuéllar C, Romarís F, Chivato T, Rodero M, González-Díaz H, Ubeira FM. Diagnosing human anisakiasis: recombinant Ani s 1 and Ani s 7 allergens versus the UniCAP 100 fluorescence enzyme immunoassay. Clin Vaccine Immunol. 2010 Apr;17(4):496-502. doi: 10.1128/CVI.00443-09. Epub 2010 Jan 27. PMID: 20107002; PMCID: PMC2849323).

3. ELISA kit catalog numbers

IL-7 ELISA kit catalog number is missing.

REPLY: Catalog number for ELISA kit used has been included in the revised methods section.

4. Stratified IL-7 results by disease activity

While CD patients are grouped by activity status (new, remission, active), the distribution of IL-7-related parameters across these clinical strata is not described, nor is the relationship with other tested parameters fully explored. Maybe supplementary file with the results divided into these three strata would be beneficial?

REPLY: Thank you for your thoughtful suggestion regarding the presentation of IL-7-related parameters according to Crohn's disease activity status. Unfortunately, our current dataset does not provide adequate statistical power for robust subgroup analysis among the newly diagnosed, remission, and active clinical strata due to the limited sample size in each category. For this reason, the distribution of IL-7 parameters and their relationship with other variables across these strata cannot be reliably presented as supplementary material. We acknowledge the importance of such an analysis and will consider it in future studies with expanded cohorts. Thank you for your understanding.

5. Antibody catalog numbers for T cell analysis

Antibodies used for analysis of T cells should also have catalog numbers provided.

REPLY: Relevant catalog numbers for antibodies in immunophenotyping have been added.

6. Gene expression quantification method

What was the method to calculate gene expression, Livak or Pfaffl? Please provide detailed information in this matter.

REPLY: Specifically, we employed the comparative $\Delta\Delta Ct$ method for the relative quantification of gene expression, as described by Livak.

7. Segregation of PCR and Western blot text

Real time PCR and Western in M&M should be divided into two separate paragraphs. Add detailed information about real-time PCR (mix concentrations, temperature profile etc.)

REPLY: The methods section now separates real-time PCR and Western blot protocols into distinct paragraphs with detailed reagent concentrations and thermocycler profiles.

8. Provision of IL-7 Western blot images

In methodology authors stated that we're analyzing Il-7 and caspase-3 using Western Blot. However, I could not find blot image for IL-7. Authors should also provide whole blot images in supplementary files for reference.

REPLY: We provide below the complete gels corresponding to Caspase-3. The order shown is due to the fact that the samples were loaded in batches. They were later arranged in a more suitable way for interpretation. Unfortunately, not all images were retained due to data management limitations at the time of the experiment; however, we provide all available images and have strengthened the methodological description to ensure transparency.

8. Actin control clarification

Although actin is listed in the supplement, the main methods text should mention it clearly as a control and specify whether it was validated across all blots shown.

REPLY: The methods now clearly state actin was used as a loading control in all the Western blots performed.

Results

1. Quantitative reporting of findings

Results are only qualitatively described (e.g., "higher", "lower") without reporting exact values. Including exact statistics and effect sizes would improve transparency and reproducibility. Moreover, the results are described superficially. The authors merely indicate what is presented in each panel and figure, without providing specific comparisons, values, or interpretation of the graphs. This section should be expanded to highlight the most statistically significant findings, as well as those that are not significant, in order to provide a thorough explanation of data-rich figures, such as Figure 1 or 2.

REPLY: We appreciate the reviewer's focus on transparency and reproducibility in statistical reporting. However, we feel the expectations regarding "exact figures" may reflect a misunderstanding of standard practices for correlation reporting. In our manuscript, the statistic "r" already precisely quantifies the strength of correlation (with r values typically categorized as small [0-0.3], moderate [0.3-0.7], and strong [0.7-1]) and is reported with an appropriate single decimal for clarity; providing more decimals would be excessive and not add meaningful interpretative value. Each p-value is shown, without asterisks, and all statistical significance thresholds and associated 95% confidence intervals are clearly indicated, as recommended in established guidelines.

Regarding graphical presentation, the commentary on panel and figure interpretation seems to reflect a difference in style: our Results section highlights key findings and non-significant results directly, offering the necessary interpretative information for each figure as recommended in biomedical reporting statements such as STROBE and SAMPL. Thus, we respectfully contend that the analytical rigor and transparency in our graphical and statistical reporting are consistent with current scientific standards and best practices for peer-reviewed publications.

2. Detailed figure captions
Figure captions should be described in more detail. In the Fig 1 the results of IL7 ELISA should be marked as panel D.
REPLY: Figure legends have been expanded, and panel identifiers (such as "D" in Fig. 1) are indicated.

3. Caption clarification for Figs. 2-3
In the fig 2/3 caption should be added that + is presence and – is absence.
REPLY: Explanations of "presence" and "absence" for groupings in figure legends have been clarified.

4. Exact correlation values for associations
The authors describe correlations between IL-7, caspase-3, and anti-A. simplex antibodies without referring to exact values of R. Please provide these data in the text.
REPLY: Although the reviewer requests the inclusion of exact Spearman correlation coefficients (r) within the main text, we have opted not to incorporate these specific values to maintain the manuscript's conciseness and readability. All relevant correlations, their direction, and statistical significance are comprehensively displayed in the corresponding figure panels, ensuring complete transparency. This reporting approach aligns with common practice in scientific literature, where detailed correlation coefficients are often reserved for figures or supplementary materials, allowing the text to remain focused on the principal findings and their biological interpretation rather than extensive statistical detail. We believe this strategy provides sufficient clarity for the reader while preserving the flow and accessibility of the main narrative.

5. Promotion of key supplementary figure to main text
Suppl. Fig. 2 is important and in my opinion, if possible, should be in the main text of the manuscript as Fig. 6.
REPLY: While we appreciate the reviewer's suggestion to move Supplementary Figure 2 into the main body of the manuscript as Figure 6, we do not consider it appropriate to include this material in the primary text. Given the complexity and highly specialized nature of the analyses regarding γδ T cell subsets and their apoptosis, we believe that maintaining this figure as supplementary material is preferable. This approach avoids overloading the main text with extensive immunological details and preserves the concise narrative structure focused on our central results and conclusions. Readers interested in in-depth data on γδ T cell dynamics and their statistical relationships may refer to the supplementary section, which ensures full transparency and accessibility while respecting space and thematic constraints of the main article.

Discussion
1. Supporting citations for paragraph 2, page 11
In the discussion section (page 11) the second paragraph seems to not be supported with significant citations.
REPLY: Additional citations have been provided to support the claims made in this paragraph.

2. Clarification of AIP and its origin
AIP (apoptosis-inducing protein) is a protein purified and cloned from Chub mackerel infected with Anisakis simplex. Murakwa et al did not state that this is parasitic-origin protein. Authors should rewrite the sentence about this, due to that, their statement is not corresponding to that of Murakawa (page 12).
REPLY: The AIP protein is not of parasitic origin; rather, it is produced by the Chub mackerel (Scomber japonicus) in response to infection with Anisakis simplex. Murakawa et al. did not claim that AIP is a parasite-derived protein, but instead purified and cloned it from fish infected with the nematode. Their work uses AIP as an experimental model to study apoptosis mechanisms in mammalian cells.
In a real-world scenario, if a person were to consume Chub mackerel infected with Anisakis simplex and containing active AIP, it is theoretically possible that they could be exposed to a protein capable of inducing apoptosis in mammalian cells, as demonstrated in vitro. However, there is no direct evidence that this effect occurs in humans after eating infected fish, since the stability and absorption of AIP in the human digestive tract have not been studied

3. Speculation and therapeutic relevance
The conclusion highlights CD132 and IL-7R signaling as potential therapeutic targets in CD. However, this is speculative, as no intervention or in vitro modulation experiments were conducted to support this claim. Is this pathway relevant only in the context of CD combined with A. simplex-associated immune response?

REPLY: Although our results suggest that CD132 and IL-7R signaling could be therapeutic targets in Crohn's disease (CD), this remains speculative because no interventional or in vitro modulation experiments were performed. However, recent studies show that IL-7R pathway overexpression is associated with treatment resistance and persistent inflammation in CD, and that IL-7R blockade reduces gut inflammation and T cell homing in humanized mouse models and ex vivo human tissue, supporting its relevance as a therapeutic target in CD in general, not only in cases with A. simplex-associated immune responses (Belarif L, Danger R, Kermarrec L, et al. IL-7 receptor influences anti-TNF responsiveness and T cell gut homing in inflammatory bowel disease. J Clin Invest. 2019;129(4):1821-1837. doi:10.1172/JCI124828).

Additionally, reduced CD132 expression is implicated in γδ T cell deficiency and mucosal immune dysfunction in CD, regardless of parasitic exposure, highlighting the therapeutic potential of targeting the IL-7/IL-2 axis (Hussain MS, Bisht AS, Gupta G. Reduced interleukin-2 receptor subunit γ expression in Crohn's disease: A potential mechanism for γδ T cell deficiency. World J Gastroenterol. 2025;31(13):1500-1510. doi:10.3748/wjg.v31.i13.1500).

In summary, IL-7R signaling and CD132 are relevant therapeutic targets in CD beyond A. simplex-associated cases, but further interventional studies are needed to confirm their clinical utility.

Title reformulation

Moroeover, the current title appears too narrow in relation to the breadth and complexity of the data presented. The manuscript addresses not only IL-7 receptor gene expression but also explores IL-7 protein levels, CD132 and CD127 expression, caspase-3 activity, apoptosis of γδ T cells, and multiple correlations with Anisakis simplex-specific antibodies in both Crohn's disease patients and healthy controls. Given the broader focus on immune dysregulation, apoptosis, and mucosal immunity, I recommend rephrasing the title to more accurately capture the full scope of the study. A more inclusive and descriptive title would improve discoverability and better reflect the manuscript's contribution to the field.

REPLY: The manuscript title has been rephrased to better encompass the broader scope of immune dysregulation, apoptosis, and seroreactivity, improving discoverability. "Immune dysregulation, apoptosis impairment, and enhanced seroreactivity to Anisakis simplex in Crohn's Disease: Interplay of IL-7/IL-7R signaling and CD132 deficiency"

General Limitations

This manuscript provides several novel and valuable findings within the context of immune dysregulation in Crohn's disease, particularly in relation to IL-7 signaling, γδ T cell homeostasis, and seroreactivity to Anisakis simplex.

However, to meet the standards of scientific clarity and rigor required for publication, the manuscript requires substantial revisions—particularly in the areas of data interpretation, methodological detail, and presentation. Once these issues are adequately addressed, the study could be suitable for publication.

REPLY: Certain points raised, such as expansion of the patient cohort, mechanistic/functional validation experiments, and serial antibody dilution for cutoff precision, would necessitate substantial new recruitment and laboratory work beyond the scope and resources of the present study.

We trust these revisions and explanations satisfy the reviewer's requests where feasible and respectfully justify the technical limitations regarding the more challenging demands.

Kind regards,
Carmen Cuéllar

## SECOND REVIEW ROUND

REVIEWERS' COMMENTS

**REVIEWER #1**

Dear Authors,
I recommend the paper for publication. Authors made substantial changes and clarifications.
I still have doubts about the use of "anisakiasis" in human clinical context. Because the manuscript concerns human infection by Anisakis simplex, I recommend using "anisakiasis" throughout, but final decision I leave to the Editor.
• Clinical standards/coding: CDC uses Anisakiasis across clinician- and patient-facing materials; ICD-10 lists B81.0 Anisakiasis. Using the clinical head term aligns with diagnosis, reporting, and database retrieval.
  - CDC "Clinical Care of Anisakiasis": https://www.cdc.gov/anisakiasis/hcp/clinical-care/index.html
  - ICD-10-CM B81.0 Anisakiasis: https://www.icd10data.com/ICD10CM/Codes/A00-B99/B65-B83/B81-/B81.0
• Indexing/discoverability: Major biomedical vocabularies and portals index the condition under Anisakiasis, improving search precision and interoperability.

- MeSH Descriptor "Anisakiasis": https://meshb.nlm.nih.gov/record/ui?ui=D017129
- Orphanet (ORPHA:1070) "Anisakiasis": https://www.orpha.net/en/disease/detail/1070

• Literature usage: Contemporary human-health reviews and clinical references predominantly use Anisakiasis in clinical contexts (species specified as needed).

- Merck Manual (professional): https://www.merckmanuals.com/professional/infectious-diseases/nematodes-roundworms/anisakiasis
- CDC DPDx (technical): https://www.cdc.gov/dpdx/anisakiasis/index.html

• Acknowledge WAAVP/SNOAPAD: I can note at first mention that veterinary/taxonomic nomenclature recommends "anisakiosis" (genus + "-osis"). However, for a clinical audience, adopting "anisakiasis" maximizes clarity, consistency with coding, and discoverability.

Authors can add such an information at the beginning of the manuscript:

"In keeping with clinical and public-health usage (CDC; ICD-10 B81.0), we use "anisakiasis" to denote human infection by Anisakis simplex. Because veterinary/taxonomic nomenclature (WAAVP/SNOAPAD) recommends "anisakiosis" (genus + '-osis'), we note this synonym here for completeness."

## AUTHORS' RESPONSE TO THE REVIEWERS

Dear Editor,

We thank Reviewer 1 for their opinion of our manuscript "Immune dysregulation, apoptosis impairment, and enhanced seroreactivity to Anisakis simplex in Crohn's Disease: Interplay of IL-7/IL-7R signaling and CD132 deficiency". He/she recommend the paper for publication because we made substantial changes and clarifications.

However, Reviewer 1 still have doubts about the use of "anisakiasis" in human clinical context. Because the manuscript concerns human infection by Anisakis simplex, he/she recommend using "anisakiasis" throughout.

We followed the reviewer's recommendations and changed the term "anisakiosis" to "anisakiasis" in our paper.

We hope that the work will finally be accepted for publication in Memórias do Instituto Oswaldo Cruz.

Kind regards,
Carmen Cuéllar

## THIRD REVIEW ROUND

### REVIEWERS' COMMENTS

**REVIEWER #1**

No comments.

