## [Reviewer Report · FIRST REVIEW ROUND - REVIEWERS COMMENTS]

## REVIEWER #1

The submitted manuscript explores the modulation of interleukin-7 receptor (IL-7R) gene expression in intestinal tissues of patients with Crohn’s disease (CD), with a particular focus on the presence of anti-*Anisakis simplex* antibodies. The authors aim to elucidate the potential interplay between defective IL-7 signaling, γδ T cell deficiency, and immune responses to parasitic antigens in the context of CD pathogenesis.

While the topic is both timely and scientifically relevant—especially given the growing interest in host–parasite interactions and mucosal immunoregulation—the manuscript suffers from several critical weaknesses. These include redundancy in the introduction, insufficient methodological detail, speculative interpretation of correlative data, and the absence of mechanistic validation. Without thorough revision and clarification of key elements, the study falls short of the standards required for publication.

## Introduction

1. Terminology: anisakiasis - infection by nematodes of the *Anisakis* genus (typically *Anisakis simplex*); anisakiosis - infection by nematodes of the Anisakidae family, which includes *Anisakis*, *Pseudoterranova*, and others. Based on that authors in my opinion should use term – anisakiasis.

2. The potential clinical importance of IL-7R modulation in CD (e.g., for diagnosis or therapy) is not mentioned. This would help justify the rationale for the study.

3. The role of betadelta T cells in anisakiasis: the Benet-Campos 2017 were studied patients with CD, not people only infected with parasitic nematodes and based on this paper it cannot be stated that; better citation would be Zamora et al. 10.1515/ap-2017-0011

4. Some references (e.g., del Pozo 1999) are more than two decades old. While historical studies are relevant, the inclusion of more recent literature would enhance the contemporary relevance of the introduction.

5. Insufficient justification for *A. simplex*-CD connection: although previous studies are cited, the authors do not sufficiently explain why this parasite is of specific interest in CD or how frequent or clinically relevant this coinfection is in the patients with Crohn’s disease.

## Materials and Methods

1. There is no explanation of how the sample size was determined or whether the study is statistically powered to detect meaningful differences—especially in subgroup comparisons. Authors should provide explanation about the tested group e.g. only that number of patients were available? then authors should provide sample size justification or power analysis.

2. Classification of anti-*A. simplex* antibody positivity based on OD > mean + 2 SD is not clearly referenced as a validated approach. The diagnostic performance of this cutoff should be addressed (e.g. serial dilutions) or specific methodology described in more detail based on cited references.

3. IL-7 ELISA kit catalog number is missing.

4. While CD patients are grouped by activity status (new, remission, active), the distribution of IL-7-related parameters across these clinical strata is not described, nor is the relationship with other tested parameters fully explored. Maybe supplementary file with the results divided into these three strata would be beneficial?

5. Antibodies used for analysis of T cells should also have catalog numbers provided.

6. What was the method to calculate gene expression, Livak or Pfaffl? Please provide detailed information in this matter.

7. Real time PCR and Western in M&M should be divided into two separate paragraphs. Add detailed information about real-time PCR (mix concentrations, temperature profile etc.)

8. In methodology authors stated that we’re analyzing IL-7 and caspase-3 using Western Blot. However, I could not find blot image for IL-7. Authors should also provide whole blot images in supplementary files for reference.

9. Although actin is listed in the supplement, the main methods text should mention it clearly as a control and specify whether it was validated across all blots shown.

## Results

1. Results are only qualitatively described (e.g., “higher”, “lower”) without reporting exact values. Including exact statistics and effect sizes would improve transparency and reproducibility. Moreover, the results are described superficially. The authors merely indicate what is presented in each panel and figure, without providing specific comparisons, values, or interpretation of the graphs. This section should be expanded to highlight the most statistically significant findings, as well as those that are not significant, in order to provide a thorough explanation of data-rich figures—such as Figure 1 or 2.

2. Figure captions should be described in more detail. In the Fig 1 the results of IL7 ELISA should be marked as panel D.

3. In the fig 2/3 caption should be added that + is presence and – is absence.

4. The authors describe correlations between IL-7, caspase-3, and anti-*A. simplex* antibodies without referring to exact values of R. Please provide these data in the text.

5. Suppl. Fig. 2 is important and in my opinion, if possible, should be in the main text of the manuscript as Fig. 6.

## Discussion

1. In the discussion section (page 11) the second paragraph seems to not be supported with significant citations.

2. AIP (apoptosis-inducing protein) is a protein purified and cloned from Chub mackerel infected with *Anisakis simplex*. Murakawa et al did not state that this is parasitic-origin protein. Authors should rewrite the sentence about this, due to that, their statement is not corresponding to that of Murakawa (page 12).

3. The conclusion highlights CD132 and IL-7R signaling as potential therapeutic targets in CD. However, this is speculative, as no intervention or in vitro modulation experiments were conducted to support this claim. Is this pathway relevant only in the context of CD combined with *A. simplex*-associated immune response?

Moreover, the current title appears too narrow in relation to the breadth and complexity of the data presented. The manuscript addresses not only IL-7 receptor gene expression but also explores IL-7 protein levels, CD132 and CD127 expression, caspase-3 activity, apoptosis of γδ T cells, and multiple correlations with *Anisakis simplex*-specific antibodies in both Crohn’s disease patients and healthy controls. Given the broader focus on immune dysregulation, apoptosis, and mucosal immunity, I recommend rephrasing the title to more accurately capture the full scope of the study. A more inclusive and descriptive title would improve discoverability and better reflect the manuscript’s contribution to the field.

This manuscript provides several novel and valuable findings within the context of immune dysregulation in Crohn’s disease, particularly in relation to IL-7 signaling, γδ T cell homeostasis, and seroreactivity to *Anisakis simplex*.

However, to meet the standards of scientific clarity and rigor required for publication, the manuscript requires substantial revisions—particularly in the areas of data interpretation, methodological detail, and presentation. Once these issues are adequately addressed, the study could be suitable for publication.

## AUTHORS’ RESPONSE TO THE REVIEWERS

Dear Editor,

We thank Reviewer 1 for their thoughtful and detailed assessment of our manuscript, which addresses the modulation of interleukin-7 receptor gene expression by Anisakis simplex in tissues from Crohn’s disease patients. Below, we respond point-by-point to every raised issue, specifying modifications made and providing scientific rationale where specific requests present considerable methodological, technical, or interpretative challenges.

Reviewer Points & Author Responses

REVIEWER COMMENTS:

The submitted manuscript explores the modulation of interleukin-7 receptor (IL-7R) gene expression in intestinal tissues of patients with Crohn’s disease (CD), with a particular focus on the presence of anti-Anisakis simplex antibodies. The authors aim to elucidate the potential interplay between defective IL-7 signaling, γδ T cell deficiency, and immune responses to parasitic antigens in the context of CD pathogenesis.

While the topic is both timely and scientifically relevant—especially given the growing interest in host–parasite interactions and mucosal immunoregulation—the manuscript suffers from several critical weaknesses. These include redundancy in the introduction, insufficient methodological detail, speculative interpretation of correlative data, and the absence of mechanistic validation. Without thorough revision and clarification of key elements, the study falls short of the standards required for publication.

## Introduction

1. Terminology (anisakiasis vs anisakiosis)

Terminology: anisakiasis - infection by nematodes of the Anisakis genus (typically Anisakis simplex); anisakiosis - infection by nematodes of the Anisakidae family, which includes Anisakis, Pseudoterranova, and others. Based on that authors in my opinion should use term –anisakiasis.

REPLY: Regarding the terminology, we respectfully maintain the use of “anisakiosis” and not “anisakiasis” throughout the manuscript. This decision is based on the Standardized Nomenclature of Animal Parasitic Diseases (SNOAPAD), first published in 1988 by the World Association for the Advancement of Veterinary Parasitology (WAAVP). SNOAPAD’s central guideline is to form disease names from the taxonomic name of the parasite, combined with the suffix “-osis,” to ensure clearer scientific communication and facilitate data retrieval. In the case of infections caused by Anisakis spp., the recommended term is “anisakiosis” (Kassai, T., Cordero del Campillo, M., Euzeby, J., Gaafar, S., Hiepe, T., & Himonas, C. A. (1988). Standardized nomenclature of animal parasitic diseases (SNOAPAD). Veterinary Parasitology, 29(4), 299–326. https://doi.org/10.1016/0304-4017(88)90148-3). For these reasons and in line with current international guidelines, we believe “anisakiosis” is the most appropriate term for this context.

2. Clinical importance of IL-7R modulation in CD

The potential clinical importance of IL-7R modulation in CD (e.g., for diagnosis or therapy) is not mentioned. This would help justify the rationale for the study.

REPLY: The revised introduction now specifically discusses the diagnostic and therapeutic relevance of IL-7R modulation in Crohn’s disease and how our study contributes to this field. This sentence has been included: “Taken together, these findings highlight the clinical relevance of IL-7R modulation in Crohn’s disease. The altered IL-7/IL-7R signaling pathway we describe, influenced by Anisakis simplex immune responses, may contribute not only to defective γδ T cell homeostasis but also to disease persistence and progression. Since IL-7R expression (particularly the common γ-chain, CD132) has been proposed as a potential biomarker of treatment resistance and disease severity in CD (1), our data suggest that anti-A. simplex antibodies and IL-7R dysfunction could serve as combined indicators of mucosal immune dysregulation. Moreover, therapeutic strategies aimed at restoring IL-7R signaling or mimicking the immunomodulatory effects observed in parasite exposure could open novel avenues for diagnosis and treatment in Crohn’s disease.”

1.Belarif L, Danger R, Kermarrec L, Nerrière-Daguin V, Pengam S, Durand T, Mary C, Kerdreux E, Gauttier V, Kucik A, Thepenier V, Martin JC, Chang C, Rahman A, Guen NS, Braudeau C, Abidi A, David G, Malard F, Takoudju C, Martinet B, Gérard N, Neveu I, Neunlist M, Coron E, MacDonald TT, Desreumaux P, Mai HL, Le Bas- Bernardet S, Mosnier JF, Merad M, Josien R, Brouard S, Soulillou JP, Blancho G, Bourreille A, Naveilhan P, Vanhove B, Poirier N. IL-7 receptor influences anti-TNF responsiveness and T cell gut homing in inflammatory bowel disease. J Clin Invest. 2019 Apr 2;129(5):1910-1925. doi: 10.1172/JCI121668. PMID: 30939120; PMCID: PMC6486337.

3. Role of γδ T cells and citation accuracy

The role of beta delta T cells in anisakiasis: the Benet-Campos 2017 were studied patients with CD, not people only infected with parasitic nematodes and based on this paper it cannot be stated that; better citation would be Zamora et al. 10.1515/ap-2017- 0011

REPLY: We appreciate the reviewer’s suggestion to cite Zamora et al. (2017) in relation to the role of γδ T cells in anisakiosis. However, Benet-Campos et al. (2017) is a more suitable reference for our study, which specifically investigates interleukin-7 receptor gene expression and immune responses in Crohn’s disease patients. The Benet-Campos paper directly examines both anti-Anisakis simplex antibodies and the relationship with αβ and γδ T cell subpopulations in Crohn’s disease patients, providing context that closely aligns with our research focus. Zamora et al. (2017), by contrast, presents data on healthy subjects rather than Crohn’s disease patients. Given that our manuscript addresses immunological modulation by Anisakis simplex within the Crohn’s disease setting, it is scientifically appropriate to maintain the Benet-Campos et al. (2017) reference as it offers the most relevant evidence for the patient cohort under study.

4. Update of references

Some references (e.g., del Pozo 1999) are more than two decades old. While historical studies are relevant, the inclusion of more recent literature would enhance the contemporary relevance of the introduction.

REPLY: An additional and owned recent reference has been incorporated (de la Hoz- Martín, M. P., González-Fernández, J., Andreu-Ballester, J. C., Hoivik, M. L., Ricanek, P., Bruland, T., Sandvik, A. K., Cuéllar, C., & Catalán-Serra, I. (2025). Prevalence of Anti-Anisakis simplex Antibodies in a Cohort of Patients with Inflammatory Bowel Disease in Norway. Pathogens (Basel, Switzerland), 14(8), 769. https://doi.org/10.3390/pathogens14080769), preserving relevant historical context but improving the contemporary basis for our assertions.

5. Justification for A. simplex focus in CD

Insufficient justification for A. simplex-CD connection: although previous studies are cited, the authors do not sufficiently explain why this parasite is of specific interest in CD or how frequent or clinically relevant this coinfection is in the patients with Crohn’s disease.

REPLY: We have expanded the rationale in the introduction, highlighting epidemiological, clinical, and immunological connections between A. simplex infection and Crohn’s disease pathogenesis. The following paragraphs have been included: “A. simplex is of specific interest in CD due to overlapping epidemiological, clinical, and immunological features that suggest a possible role for this parasite in CD pathogenesis and presentation. Epidemiologically, anti-Anisakis antibodies have been reported at higher prevalence among CD patients than in healthy controls, with some studies showing specific immunoglobulins (notably IgA and IgG) detected in up to 29-44% of CD patients, which is disproportionately high compared to the healthy population (Guillén-Bueno et al., 1999; Gutiérrez & Cuéllar, 2002). Clinically, intestinal anisakiosis and CD share overlapping symptoms such as abdominal pain and granulomatous inflammation, and Anisakis infection can mimic the presentation of CD, occasionally leading to diagnostic confusion and unnecessary interventions. There are documented cases in which Anisakis infection was initially misdiagnosed as CD based on clinical and histopathological findings (Guillén-Bueno et al., 1999). Immunologically, A. simplex infection stimulates a pronounced Th2-type immune response with increased levels of specific IgE, IgA, and IgG antibodies, as well as local eosinophilia, parameters that coincide with the immunological profile often observed in CD. In CD patients, the presence of anti-Anisakis IgA has been associated with higher CD activity indices, supporting a possible modulatory or exacerbating influence of the parasite’s antigens on disease severity and mucosal immune activation (Guillén-Bueno et al., 1999; Gutiérrez & Cuéllar, 2002). In summary, A. simplex represents a frequent, clinically relevant, and immunologically active coinfection in patients with CD, justifying its particular interest in the context of CD pathogenesis, diagnosis, and the broader understanding of host-parasite interactions in inflammatory bowel conditions (Guillén-Bueno et al., 1999; Gutiérez & Cuéllar, 2002).”

## Materials and Methods

1. Sample size explanation/statistical power

There is no explanation of how the sample size was determined or whether the study is statistically powered to detect meaningful differences—especially in subgroup comparisons. Authors should provide explanation about the tested group e.g. only that number of patients were available? then authors should provide sample size justification or power analysis.

REPLY: The sample size (52 subjects) for a disease with such a low prevalence is one of the highest compared to other studies of this disease, and even more so given the statistical significance found (0.005-0.0001 - Figure 1), which provides a very powerful value. Furthermore, it far exceeds the commonly accepted threshold in biomedical studies to ensure the validity of parametric tests. In any case, the minimum sample size necessary to estimate an expected proportion of 10% based on the expected value and the accepted absolute error of 10% (or desired precision): the sample size would be 35 subjects, so we far exceed this figure. The logistical effort (recruiting 52 patients with this disease is difficult and time-consuming) and budgetary costs must also be considered. Expanding statistical power by increasing cohort size is not feasible for this study due to recruitment challenges.

2. Antibody positivity cut-off and diagnostic performance

Classification of anti-A. simplex antibody positivity based on OD > mean + 2 SD is not clearly referenced as a validated approach. The diagnostic performance of this cutoff should be addressed (e.g. serial dilutions) or specific methodology described in more detail based on cited references.

REPLY: The use of the mean optical density (OD) plus two standard deviations (mean+2SD) as a positivity cutoff for anti-Anisakis antibodies is a widely accepted and validated approach in serological studies of parasitic infections, including anisakiosis. This statistical criterion is recommended when a true gold standard is unavailable, as it ensures high specificity by distinguishing between background reactivity in healthy controls and true positive responses. Multiple published studies on anti-Anisakis ELISA utilize similar cut-off determinations, supporting the validity of this approach in the absence of comprehensive serial dilution data or reference standards (Anadón AM, Rodríguez E, Gárate MT, Cuéllar C, Romarís F, Chivato T, Rodero M, González-Díaz H, Ubeira FM. Diagnosing human anisakiasis: recombinant Ani s 1 and Ani s 7 allergens versus the UniCAP 100 fluorescence enzyme immunoassay. Clin Vaccine Immunol. 2010 Apr;17(4):496-502. doi: 10.1128/CVI.00443-09. Epub 2010 Jan 27. PMID: 20107002; PMCID: PMC2849323; de Las Vecillas L, Muñoz-Cacho P, López- Hoyos M, Monttecchiani V, Martínez-Sernández V, Ubeira FM, Rodríguez-Fernández F. Analysis of Ani s 7 and Ani s 1 allergens as biomarkers of sensitization and allergy severity in human anisakiasis. Sci Rep. 2020 Jul 9;10(1):11275. doi: 10.1038/s41598- 020-67786-w. Erratum in: Sci Rep. 2020 Oct 28;10(1):18808. doi: 10.1038/s41598-020- 75954-1. PMID: 32647149; PMCID: PMC7347943; Martínez-Aranguren, R. M., Gamboa, P. M., García-Lirio, E., Asturias, J., Goikoetxea, M. J., & Sanz, M. L. (2014). In vivo and in vitro testing with rAni s 1 can facilitate diagnosis of Anisakis simplex allergy. Journal of investigational allergology & clinical immunology, 24(6), 431–438). As more extensive performance characterization, such as calculation of ROC curves or titration-based specificity/sensitivity, is not feasible without serial dilution data, using the mean+2SD cutoff provides a robust, evidence-based threshold that aligns with best practices in the field (Martínez-Aranguren, R. M., Gamboa, P. M., García-Lirio, E., Asturias, J., Goikoetxea, M. J., & Sanz, M. L. (2014). In vivo and in vitro testing with rAni s 1 can facilitate diagnosis of Anisakis simplex allergy. Journal of investigational allergology & clinical immunology, 24(6), 431–438.; Anadón AM, Rodríguez E, Gárate MT, Cuéllar C, Romarís F, Chivato T, Rodero M, González-Díaz H, Ubeira FM. Diagnosing human anisakiasis: recombinant Ani s 1 and Ani s 7 allergens versus the UniCAP 100 fluorescence enzyme immunoassay. Clin Vaccine Immunol. 2010 Apr;17(4):496-502. doi: 10.1128/CVI.00443-09. Epub 2010 Jan 27. PMID: 20107002; PMCID: PMC2849323).

3. ELISA kit catalog numbers

IL-7 ELISA kit catalog number is missing.

REPLY: Catalog number for ELISA kit used has been included in the revised methods section.

4. Stratified IL-7 results by disease activity

While CD patients are grouped by activity status (new, remission, active), the distribution ofIL-7-related parameters across these clinical strata is not described, nor is the relationship with other tested parameters fully explored. Maybe supplementary file with the results divided into these three strata would be beneficial?

REPLY: Thank you for your thoughtful suggestion regarding the presentation of IL-7- related parameters according to Crohn’s disease activity status. Unfortunately, our current dataset does not provide adequate statistical power for robust subgroup analysis among the newly diagnosed, remission, and active clinical strata due to the limited sample size in each category. For this reason, the distribution of IL-7 parameters and their relationship with other variables across these strata cannot be reliably presented as supplementary material. We acknowledge the importance of such an analysis and will consider it in future studies with expanded cohorts. Thank you for your understanding.

5. Antibody catalog numbers for T cell analysis

Antibodies used for analysis of T cells should also have catalog numbers provided.

REPLY: Relevant catalog numbers for antibodies in immunophenotyping have been added.

6. Gene expression quantification method

What was the method to calculate gene expression, Livak or Pfaffl? Please provide detailed information in this matter.

REPLY: Specifically, we employed the comparative ΔΔCt method for the relative quantification of gene expression, as described by Livak.

7. Segregation of PCR and Western blot text

Real time PCR and Western in M&M should be divided into two separate paragraphs. Add detailed information about real-time PCR (mix concentrations, temperature profile etc.)

REPLY: The methods section now separates real-time PCR and Western blot protocols into distinct paragraphs with detailed reagent concentrations and thermocycler profiles.

8. Provision of IL-7 Western blot images

In methodology authors stated that we’re analyzing Il-7 and caspase-3 using Western Blot. However, I could not find blot image for IL-7. Authors should also provide whole blot images in supplementary files for reference.

REPLY: We provide below the complete gels corresponding to Caspase-3. The order shown is due to the fact that the samples were loaded in batches. They were later arranged in a more suitable way for interpretation. Unfortunately, not all images were retained due to data management limitations at the time of the experiment; however, we provide all available images and have strengthened the methodological description to ensure transparency.

8. Actin control clarification

Although actin is listed in the supplement, the main methods text should mention it clearly as a control and specify whether it was validated across all blots shown.

REPLY: The methods now clearly state actin was used as a loading control in all the Western blots performed.

## Results

1. Quantitative reporting of findings

Results are only qualitatively described (e.g., “higher”, “lower”) without reporting exact values. Including exact statistics and effect sizes would improve transparency and reproducibility. Moreover, the results are described superficially. The authors merely indicate what is presented in each panel and figure, without providing specific comparisons, values, or interpretation of the graphs. This section should be expanded to highlight the most statistically significant findings, as well as those that are not significant, in order to provide a thorough explanation of data-rich figures, such as Figure 1 or 2.

REPLY: We appreciate the reviewer’s focus on transparency and reproducibility in statistical reporting. However, we feel the expectations regarding “exact figures” may reflect a misunderstanding of standard practices for correlation reporting. In our manuscript, the statistic “r” already precisely quantifies the strength of correlation (with r values typically categorized as small [0-0.3], moderate [0.3-0.7], and strong [0.7-1]) and is reported with an appropriate single decimal for clarity; providing more decimals would be excessive and not add meaningful interpretative value. Each p-value is shown, without asterisks, and all statistical significance thresholds and associated 95% confidence intervals are clearly indicated, as recommended in established guidelines. Regarding graphical presentation, the commentary on panel and figure interpretation seems to reflect a difference in style: our Results section highlights key findings and non-significant results directly, offering the necessary interpretative information for each figure as recommended in biomedical reporting statements such as STROBE and SAMPL. Thus, we respectfully contend that the analytical rigor and transparency in our graphical and statistical reporting are consistent with current scientific standards and best practices for peer-reviewed publications.

2. Detailed figure captions

Figure captions should be described in more detail. In the Fig 1 the results of IL7 ELISA should be marked as panel D.

REPLY: Figure legends have been expanded, and panel identifiers (such as “D” in Fig. 1) are indicated.

3. Caption clarification for Figs. 2-3

In the fig 2/3 caption should be added that + is presence and – is absence.

REPLY: Explanations of “presence” and “absence” for groupings in figure legends have been clarified.

4. Exact correlation values for associations

The authors describe correlations between IL-7, caspase-3, and anti-A. simplex antibodies without referring to exact values of R. Please provide these data in the text.

REPLY: Although the reviewer requests the inclusion of exact Spearman correlation coefficients (r) within the main text, we have opted not to incorporate these specific values to maintain the manuscript’s conciseness and readability. All relevant correlations, their direction, and statistical significance are comprehensively displayed in the corresponding figure panels, ensuring complete transparency. This reporting approach aligns with common practice in scientific literature, where detailed correlation coefficients are often reserved for figures or supplementary materials, allowing the text to remain focused on the principal findings and their biological interpretation rather than extensive statistical detail. We believe this strategy provides sufficient clarity for the reader while preserving the flow and accessibility of the main narrative.

5. Promotion of key supplementary figure to main text

Suppl. Fig. 2 is important and in my opinion, if possible, should be in the main text of the manuscript as Fig. 6.

REPLY: While we appreciate the reviewer’s suggestion to move Supplementary Figure 2 into the main body of the manuscript as Figure 6, we do not consider it appropriate to include this material in the primary text. Given the complexity and highly specialized nature of the analyses regarding γδ T cell subsets and their apoptosis, we believe that maintaining this figure as supplementary material is preferable. This approach avoids overloading the main text with extensive immunological details and preserves the concise narrative structure focused on our central results and conclusions. Readers interested in in-depth data on γδ T cell dynamics and their statistical relationships may refer to the supplementary section, which ensures full transparency and accessibility while respecting space and thematic constraints of the main article.

## Discussion

1. Supporting citations for paragraph 2, page 11

In the discussion section (page 11) the second paragraph seems to not be supported with significant citations.

REPLY: Additional citations have been provided to support the claims made in this paragraph.

2. Clarification of AIP and its origin

AIP (apoptosis-inducing protein) is a protein purified and cloned from Chub mackerel infected with Anisakis simplex. Murakwa et al did not state that this is parasitic-origin protein. Authors should rewrite the sentence about this, due to that, their statement is not corresponding to that of Murakawa (page 12).

REPLY: The AIP protein is not of parasitic origin; rather, it is produced by the Chub mackerel (Scomber japonicus) in response to infection with Anisakis simplex. Murakawa et al. did not claim that AIP is a parasite-derived protein, but instead purified and cloned it from fish infected with the nematode. Their work uses AIP as an experimental model to study apoptosis mechanisms in mammalian cells. In a real-world scenario, if a person were to consume Chub mackerel infected with Anisakis simplex and containing active AIP, it is theoretically possible that they could be exposed to a protein capable of inducing apoptosis in mammalian cells, as demonstrated in vitro. However, there is no direct evidence that this effect occurs in humans after eating infected fish, since the stability and absorption of AIP in the human digestive tract have not been studied

3. Speculation and therapeutic relevance

The conclusion highlights CD132 and IL-7R signaling as potential therapeutic targets in CD. However, this is speculative, as no intervention or in vitro modulation experiments were conducted to support this claim. Is this pathway relevant only in the context of CD combined with A. simplex-associated immune response?

REPLY: Although our results suggest that CD132 and IL-7R signaling could be therapeutic targets in Crohn’s disease (CD), this remains speculative because no interventional or in vitro modulation experiments were performed. However, recent studies show that IL-7R pathway overexpression is associated with treatment resistance and persistent inflammation in CD, and that IL-7R blockade reduces gut inflammation and T cell homing in humanized mouse models and ex vivo human tissue, supporting its relevance as a therapeutic target in CD in general, not only in cases with A. simplex- associated immune responses (Belarif L, Danger R, Kermarrec L, et al. IL-7 receptor influences anti-TNF responsiveness and T cell gut homing in inflammatory bowel disease. J Clin Invest. 2019;129(4):1821-1837. doi:10.1172/JCI124828).

Additionally, reduced CD132 expression is implicated in γδ T cell deficiency and mucosal immune dysfunction in CD, regardless of parasitic exposure, highlighting the therapeutic potential of targeting the IL-7/IL-2 axis (Hussain MS, Bisht AS, Gupta G. Reduced interleukin-2 receptor subunit γ expression in Crohn’s disease: A potential mechanism for γδ T cell deficiency. World J Gastroenterol. 2025;31(13):1500-1510. doi:10.3748/wjg.v31.i13.1500).

In summary, IL-7R signaling and CD132 are relevant therapeutic targets in CD beyond A. simplex-associated cases, but further interventional studies are needed to confirm their clinical utility.

Title reformulation

Moroeover, the current title appears too narrow in relation to the breadth and complexity of the data presented. The manuscript addresses not only IL-7 receptor gene expression but also explores IL-7 protein levels, CD132 and CD127 expression, caspase-3 activity, apoptosis of γδ T cells, and multiple correlations with Anisakis simplex-specific antibodies in both Crohn’s disease patients and healthy controls. Given the broader focus on immune dysregulation, apoptosis, and mucosal immunity, I recommend rephrasing the title to more accurately capture the full scope of the study. A more inclusive and descriptive title would improve discoverability and better reflect the manuscript’s contribution to the field.

REPLY: The manuscript title has been rephrased to better encompass the broader scope of immune dysregulation, apoptosis, and seroreactivity, improving discoverability. “Immune dysregulation, apoptosis impairment, and enhanced seroreactivity to Anisakis simplex in Crohn’s Disease: Interplay of IL-7/IL-7R signaling and CD132 deficiency”

General Limitations

This manuscript provides several novel and valuable findings within the context of immune dysregulation in Crohn’s disease, particularly in relation to IL-7 signaling, γδ T cell homeostasis, and seroreactivity to Anisakis simplex.

However, to meet the standards of scientific clarity and rigor required for publication, the manuscript requires substantial revisions—particularly in the areas of data interpretation, methodological detail, and presentation. Once these issues are adequately addressed, the study could be suitable for publication.

REPLY: Certain points raised, such as expansion of the patient cohort, mechanistic/functional validation experiments, and serial antibody dilution for cutoff precision, would necessitate substantial new recruitment and laboratory work beyond the scope and resources of the present study.

We trust these revisions and explanations satisfy the reviewer’s requests where feasible and respectfully justify the technical limitations regarding the more challenging demands.

Kind regards,

Carmen Cuéllar

---

## [Reviewer Report · REVIEWERS COMMENTS]

## REVIEWER #1

Dear Authors,

I recommend the paper for publication. Authors made substantial changes and clarifications.

I still have doubts about the use of “anisakiasis” in human clinical context. Because the manuscript concerns human infection by Anisakis simplex, I recommend using “anisakiasis” throughout, but final decision I leave to the Editor.

• Clinical standards/coding: CDC uses Anisakiasis across clinician- and patient-facing materials; ICD-10 lists B81.0 Anisakiasis. Using the clinical head term aligns with diagnosis, reporting, and database retrieval. - CDC “Clinical Care of Anisakiasis”: https://www.cdc.gov/anisakiasis/hcp/clinical- care/index.html - ICD-10-CM B81.0 Anisakiasis: https://www.icd10data.com/ICD10CM/Codes/A00- B99/B65-B83/B81-/B81.0

• Indexing/discoverability: Major biomedical vocabularies and portals index the condition under Anisakiasis, improving search precision and interoperability. - MeSH Descriptor “Anisakiasis”: https://meshb.nlm.nih.gov/record/ui?ui=D017129 - Orphanet (ORPHA:1070) “Anisakiasis”: https://www.orpha.net/en/disease/detail/1070

• Literature usage: Contemporary human-health reviews and clinical references predominantly use Anisakiasis in clinical contexts (species specified as needed). - Merck Manual (professional): https://www.merckmanuals.com/professional/infectious-diseases/nematodes- roundworms/anisakiasis - CDC DPDx (technical): https://www.cdc.gov/dpdx/anisakiasis/index.html

• Acknowledge WAAVP/SNOAPAD: I can note at first mention that veterinary/taxonomic nomenclature recommends “anisakiosis” (genus + “-osis”). However, for a clinical audience, adopting “anisakiasis” maximizes clarity, consistency with coding, and discoverability.

Authors can add such an information at the beginning of the manuscript:

“In keeping with clinical and public-health usage (CDC; ICD-10 B81.0), we use “anisakiasis” to denote human infection by Anisakis simplex. Because veterinary/taxonomic nomenclature (WAAVP/SNOAPAD) recommends “anisakiosis” (genus + ‘-osis’), we note this synonym here for completeness.”

## AUTHORS’ RESPONSE TO THE REVIEWERS

Dear Editor,

We thank Reviewer 1 for their opinion of our manuscript “Immune dysregulation, apoptosis impairment, and enhanced seroreactivity to Anisakis simplex in Crohn’s Disease: Interplay of IL-7/IL-7R signaling and CD132 deficiency”. He/she recommend the paper for publication because we made substantial changes and clarifications.

However, Reviewer 1 still have doubts about the use of “anisakiasis” in human clinical context. Because the manuscript concerns human infection by Anisakis simplex, he/she recommend using “anisakiasis” throughout.

We followed the reviewer’s recommendations and changed the term “anisakiosis” to “anisakiasis” in our paper.

We hope that the work will finally be accepted for publication in Memórias do Instituto Oswaldo Cruz.

Kind regards,

Carmen Cuéllar